# AI-based mobile application to fight antibiotic resistance

Marco Pascucci [1,2,3,12], Guilhem Royer [4,5,6,12], Jakub Adamek[7], Mai Al Asmar[8], David Aristizabal[7], Laetitia Blanche[1], Amine Bezzarga[1,9], Guillaume Boniface-Chang[7], Alex Brunner[7], Christian Curel[10], Gabriel Dulac-Arnold[11], Rasheed M. Fakhri[8], Nada Malou[1✉], Clara Nordon[1], Vincent Runge[2], Franck Samson[2], Ellen Sebastian[7], Dena Soukieh[7], Jean-Philippe Vert[11], Christophe Ambroise[2,13✉] & Mohammed-Amin Madoui [5,13✉]

Antimicrobial resistance is a major global health threat and its development is promoted by antibiotic misuse. While disk diffusion antibiotic susceptibility testing (AST, also called antibiogram) is broadly used to test for antibiotic resistance in bacterial infections, it faces strong criticism because of inter-operator variability and the complexity of interpretative reading. Automatic reading systems address these issues, but are not always adapted or available to resource-limited settings. We present an artificial intelligence (AI)-based, offline smartphone application for antibiogram analysis. The application captures images with the phone's camera, and the user is guided throughout the analysis on the same device by a user-friendly graphical interface. An embedded expert system validates the coherence of the antibiogram data and provides interpreted results. The fully automatic measurement procedure of our application's reading system achieves an overall agreement of 90% on susceptibility categorization against a hospital-standard automatic system and 98% against manual measurement (gold standard), with reduced inter-operator variability. The application's performance showed that the automatic reading of antibiotic resistance testing is entirely feasible on a smartphone. Moreover our application is suited for resource-limited settings, and therefore has the potential to significantly increase patients' access to AST worldwide.

[1] The MSF Foundation, Paris, France. [2] Université Paris-Saclay, CNRS, Univ Evry, Laboratoire de Mathématiques et Modélisation d'Evry, 91037 Evry-Courcouronnes, France. [3] Université Paris-Saclay, CEA, CNRS, Neurospin, Baobab, Gif-sur-Yvette, France. [4] Université de Paris, IAME, UMR1137, INSERM, Paris, France. [5] Université Paris-Saclay, Univ Evry, CNRS, CEA, Génomique métabolique, 91037 Evry-Courcouronnes, France. [6] Département de prévention, diagnostic et traitement des infections, Hôpital Henri Mondor, AP-HP, Créteil, France. [7] Google.org, https://www.google.org, USA. [8] MSF Amman Hospital, Amman, Jordan. [9] X-Squad, Paris, France. [10] i2a, Montpellier, France. [11] Google Research, Brain Team, Paris, France. [12] These authors contributed equally: Marco Pascucci, Guilhem Royer. [13] These authors jointly supervised this work: Christophe Ambroise, Mohammed-Amin Madoui. ✉email: nada.malou@paris.msf.org; christophe.ambroise@univ-evry.fr; amadoui@genoscope.cns.fr

The development of new antimicrobial agents is currently outpaced by the emergence of new antimicrobial resistance[1] (AMR). The appearance and diffusion of AMR have become a serious health threat[2], whose magnitude is not yet fully understood because of the lack of data, especially in areas where the access to antimicrobial susceptibility testing is difficult. A high-profile review[3] forecasts ten million deaths worldwide by 2050. Although these numbers have been criticized[2], these studies underline the critical health burden of AMR and the need for global data[2,4].

Testing the susceptibility of bacteria is important for patient treatment and, if done systematically, gathering data can provide precious epidemiological information. Different test methods[5] exist. Arguably the most widely used is the Kirby–Bauer disk diffusion test.

In this test, cellulose disks (pellets) containing antibiotics at a given concentration are placed in a Petri dish with an agar-based growth medium previously inoculated with bacteria. While the plate is left to incubate, the antibiotic diffuses from the pellet into the agar. The antibiotic concentration is highest near a pellet and decreases radially as the distance from the disk increases[6]. The bacteria cannot grow around those disks that contain antibiotics to which they are susceptible. The growth of the bacterial colony stops at a distance from the pellet which corresponds to a critical antibiotic concentration, forming a visible bacteria-free area around the cellulose disk. This is called a *zone of inhibition*. After incubation, the diameter of the zone of inhibition around each antibiotic disk is measured: the categorization of the microorganisms as susceptible (S) Intermediate (I) or Resistant (R) is obtained by comparison of the diameter against standard breakpoints[7] established by international committees such as the European Committee on Antimicrobial Susceptibility Testing (EUCAST) or the Clinical and Laboratory Standards Institute (CLSI)[8].

The disk diffusion method is relatively simple, can be performed entirely by hand, requires no advanced hardware, and has a low cost. However, it is criticized for several reasons. First, it is labor intensive and time consuming. Second, it is subject to important inter-operator variability: accurate performance of disk diffusion testing relies on proficient technicians, starting with the quality of plate preparation (e.g., inoculum, purity)[9]. The diameter of the inhibition zone is measured by eye with a caliper or ruler and approximated to the closest millimeter[10]. However, the inhibition zone might not be a perfect disk (e.g., if the inhibition zones overlap) or if the pellet is too close to the border of the dish. In this case, the problem of measuring a diameter is ill-posed and, together with intrinsic measurement error, introduces subjectivity and inter-operator variability in the measurement. Third, it requires an advanced level of expertise for interpretation. Sometimes, the inhibition zone diameter is not sufficient by itself to determine the susceptibility. Indeed, several mechanisms of resistance are expressed at a low level in vitro but have a major impact in vivo and can lead to treatment failure. Moreover, susceptibility to a whole class of antibiotics or a given molecule class can sometimes be inferred from the susceptibility to another one, thus reducing the number of required tests. In those cases, interpretative reading is needed. Interpretation is based on expert rules published and updated by scientific societies, such as EUCAST in Europe[11].

Automatic reading systems have been introduced to alleviate the drawbacks of disk diffusion AST[12,13]. These systems acquire pictures of the plate and automatically measure the diameters of the inhibition zones. Most of them include an *expert system* that can elaborate interpreted results. It helps mitigate the risk that the laboratory reports erroneous susceptibility results and ensures compliance with regulatory guidelines. Commercial devices[14–16] that automatically read antibiograms are commonly used in hospitals and laboratories, but the procedures they use are not fully disclosed. These systems aim towards great and flawless automation and a high degree of standardization of the culturing procedures in order to concurrently increase quality and turn-around times. These needs are not the same in resources-limited hospitals, where AST might be not implemented at all.

Because of their price, and hardware and infrastructure requirements, these systems are not suited to environments such as dispensaries or hospitals in resource-limited settings. Affordable solutions are few. Image processing algorithms for automatic measuring inhibition diameters have been published[17–21]. Among these, only AntibiogramJ[21] presents a fully functional user-friendly software, but it operates on a desktop computer onto which the images need to be previously transferred. We believe that reducing hardware requirements to just a smartphone is key for the adoption and diffusion of such a tool. Moreover, smartphone applications are easy to adopt and use if they follow established design patterns, and they benefit from an ecosystem that facilitates setup and updates.

This paper introduces a fully offline mobile application (the App hereafter) capable of analyzing disk diffusion ASTs and yielding interpreted results, operating entirely on a smartphone. The need of such an application was identified by Medecins Sans Frontieres (MSF), who often operates in low and middle income countries (LMIC) where AST is difficult or impossible to implement. The MSF Foundation brought together the people and skills needed for this application to be developed, truly believing that the App can have a great impact on the fields where MSF operates and the global fight against AMR.

The App combines original algorithms, using machine learning (ML) and image processing, with a rule-based expert system, for automatic AST analysis (see Fig. 1). It embeds a clinically tested third-party expert system[13,15] which could compensate for a lack of microbiology expertise. The user is guided throughout the whole analysis and can interact at any step with the user-friendly graphical interface of the application to verify and possibly correct the automatic measurements if needed. The whole analysis takes place on the same smartphone used to acquire the picture of the AST. Since it does not require any hardware other than a basic Android smartphone, and because it works completely offline (without internet connection), the App is suited for resource-limited settings. Therefore, the App could help fill the digital gap, increase patients' access to AST worldwide and possibly facilitate the collection of epidemiological data on antimicrobial resistances, the lack of which is recognized today as a major health danger[2,4].

In fact, the main aim of this application is to facilitate the adoption of the disk diffusion AST in resources-limited hospitals and laboratories where this test is not available yet. The App pursues this objective by partially alleviating the need of expert human resources, making the reading more reliable, and providing interpreted results. Therefore this application does not want to compete with high-end commercial systems, which can count on dedicated hardware. Nevertheless, in order to be reliable, it is fundamental that the App fulfills the minimum viable performance requirements, as we show in this work.

In the following, we demonstrate that the App's performance is similar to that obtained with a commercial system and conform to manual reading (considered as the gold standard[10]). The application's full automatic procedure is evaluated on antibiograms prepared in laboratory conditions both on standard and blood-enriched agar. Moreover, we explore the feasibility of an ML-based automatic detection of resistance mechanisms recognizable by peculiar shapes of the inhibition zones.

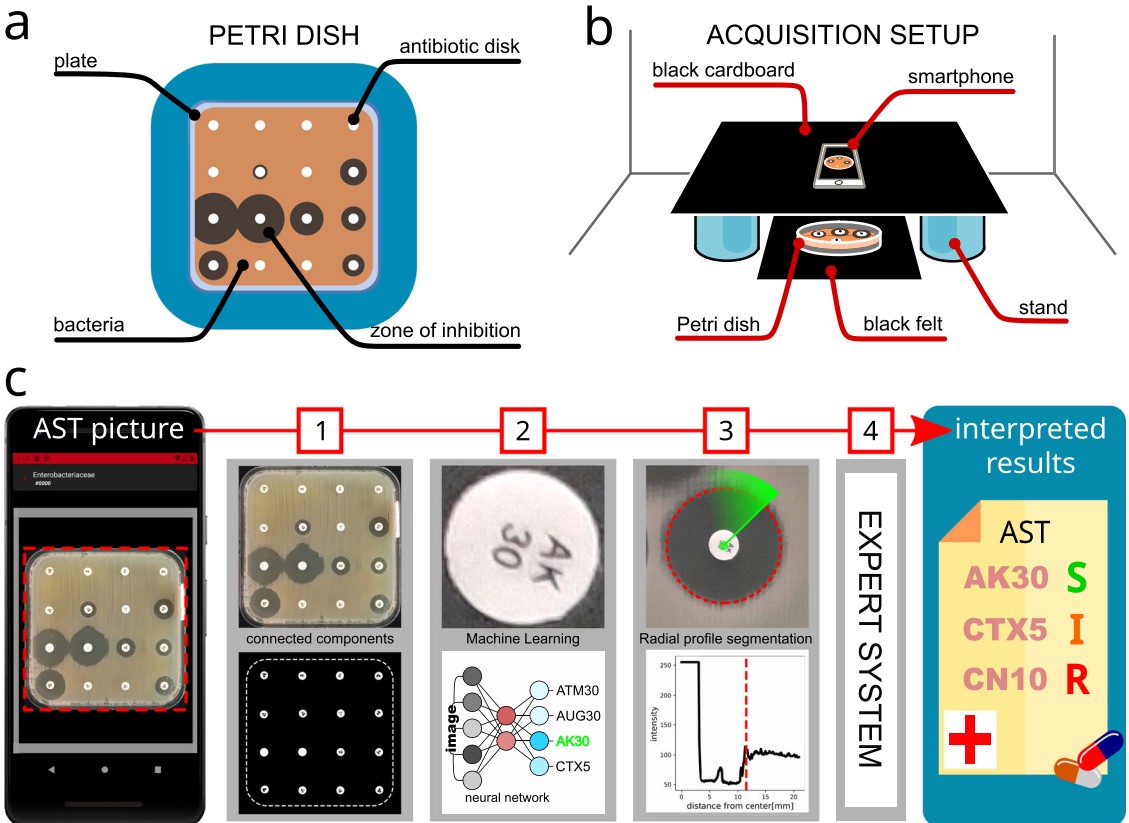

**Fig. 1 Analysis of an AST plate with the App.** A prepared and incubated Petri dish (**a**) is positioned in a simple image acquisition setup made of cardboard (**b**), we used two containers available in the laboratory as stands. A picture of the plate is taken with a smartphone and the analysis follows the workflow described in (**c**): the Petri dish image is cropped and the antibiotic disks are found (c1); the image of each antibiotic disk is fed to a ML model that identifies the antibiotic (c2); the diameter of the inhibition zone is measured (c3) with an original algorithm. Finally, the Expert System uses the diameters to output interpreted results (c4).

## Results

**Image processing steps**. The App presented in this paper is an automatic AST reading system capable of running the whole analysis of a disk-diffusion antibiogram offline on mobile devices, from image acquisition to interpreted results. It helps laboratory technicians throughout the whole analysis process, suggesting measurements, results, and interpretations. The App can be summarized in three major components: first, a dedicated image processing module (IP) that reads and analyzes the AST image; second, an expert system (ES) responsible for the interpretation of the data extracted by IP; third, a Graphical User Interface (GUI) that allows the execution of IP and ES on a smartphone, and user interaction.

The application's image processing library[22] implements an original algorithm for the measurement of the inhibition diameters (described in Methods) and uses ML for the identification of the antibiotic disks, which is unprecedented in this kind of application. The IP module consists in a C++ library developed on OpenCV[23] and Tensorflow[24]. The choice of C++ makes our library exploitable in various contexts, including desktop computers and Android and iOS mobile devices. Moreover, the library has a Python wrapper, useful for application prototyping, image batch processing, and benchmarking.

The App includes an expert system capable of performing coherence checks on the raw susceptibility and providing interpreted results, with extrapolation on non-tested antibiotics and clinical commentaries. The expert system's knowledge base is provided and regularly updated by i2a (Montpellier, France)[13] and based on up-to-date EUCAST expert rules[11]. The expert

system's engine was completely developed in TypeScript and works completely offline within the App.

Commercial AST reading systems use built-in image acquisition devices (cameras and scanners) to ensure input consistency. The App works on images of ASTs taken directly with the phone's camera, with no additional external acquisition hardware. This inevitably introduces a certain variability in the image quality. We tackle this issue by introducing a simple set of guidelines for image acquisition (see Supplementary Note 1). These guidelines are designed to optimize image quality, and therefore to reduce the need of heavy post-processing and the risk of numerical artifacts. For the same reason, perspective distortions are not corrected. Instead, we developed a simple acquisition setup (Fig. 1) which ensures parallelism between the dish and the camera's image plane. The acquisition guidelines are conceived to be inexpensive and easy to implement and integrate in the laboratory routine. Since smartphone cameras are not designed for quantitative measurements, we provide a simple method to assess the camera's optical distortions with a numerically generated AST image. Moreover, while taking the picture, the application uses the device's gyroscopes and accelerometer, if available, to force the device orientation (parallel to ground, to avoid perspective distortions) and stability (to avoid motion blur). The application also displays a visual frame that helps center the Petri dish in the picture. Although Petri dishes have standard shapes (square or disk), we do not rely on this assumption for the analysis.

The IP module analyzes an AST picture in three different sequential stages: plate cropping, detection of antibiotic disks, and

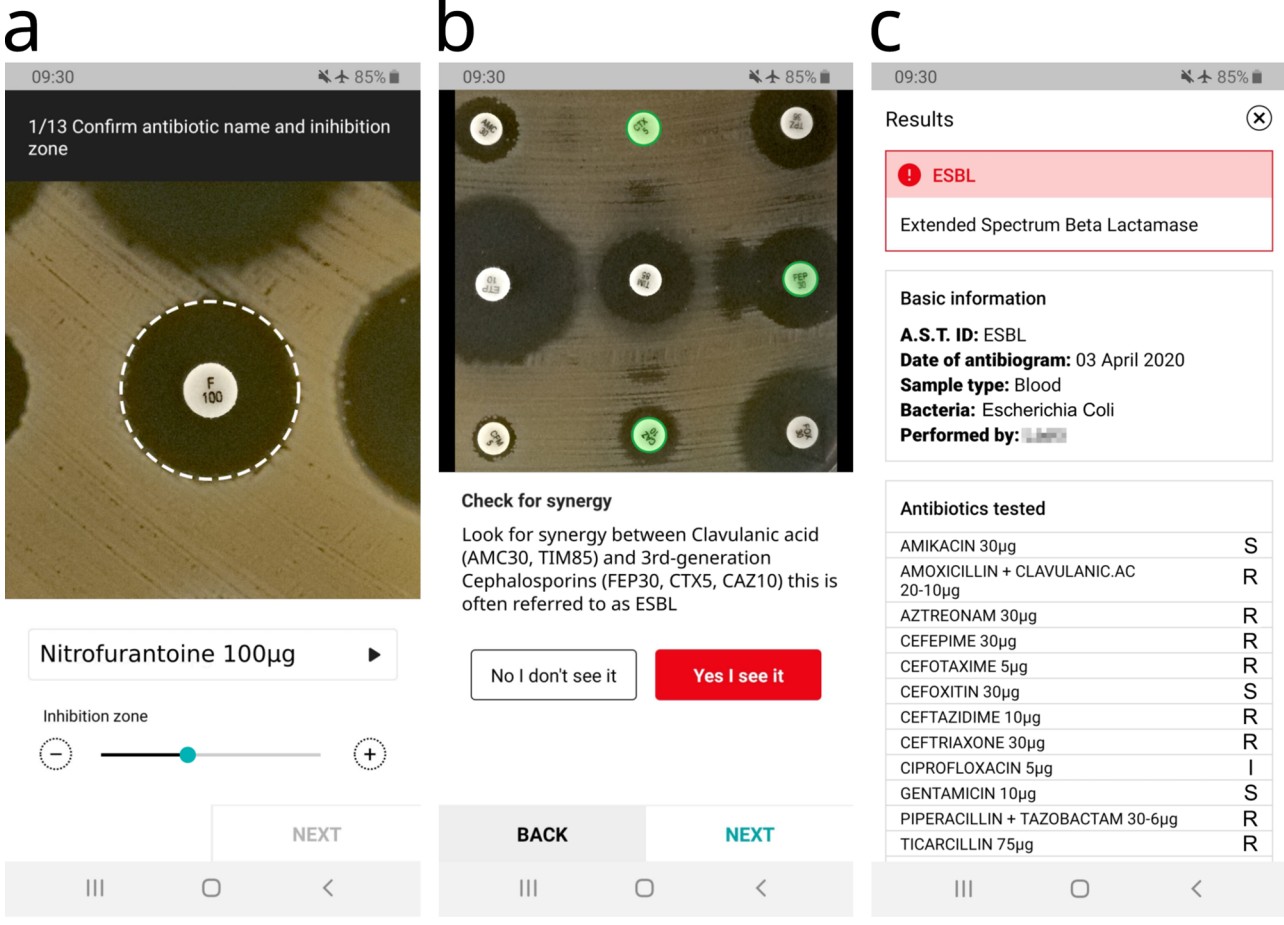

**Fig. 2 Screenshots of the App in action. a** The App displays a zoomed image of an inhibition zone and indicates with a dashed circle the automatically measured diameter and the detected antibiotic. The user can edit the results with the controls below the image. **b** The application can ask the users if they see the peculiar shapes of inhibition zones associated with certain resistance mechanisms. **c** At the end of the analysis, the interpreted results are shown to the user.

**Table 1 Description of the AST sets used for the performance evaluation of the automatic reading. For each dataset, the columns indicate: the number of single Petri dishes in the dataset, the corresponding total number of antibiotic disks, the type of growth medium, the shape of the plates, the number of independent raters measuring the diameters, the reference used as control diameter.**

| AST set | Plates | Antibiotics | Growth medium | Petri dish shape | Raters | control |
|---|---|---|---|---|---|---|
| A1 | 570 | 8168 | MH | square | 1 | SIRscan |
| A2 | 75 | 649 | blood | mixed | 1 | SIRscan |
| A3 | 8 | 98 | MH | circular | 8 | manual |

inhibition diameter measurement. The resulting diameters are used to categorize the susceptibility of the bacteria and interpret the results. These stages are described in Methods and summarized in Fig. 1. Once the antibiogram picture is taken, the Petri dish is cropped out, the antibiotic disks are found and classified according to their label, the diameter of the inhibition zones is measured and translated into susceptibility results. At this point, the expert system verifies the coherence of the measurements to highlight possible errors and finally returns the interpreted results. Some screenshots of the App are shown in Fig. 2 and a complete video demo is available online (see Supplementary Video 1).

**Preparation of antibiotic susceptibility tests**. In order to evaluate the App's performances, we ran the fully automatic analysis

procedure (without any manual intervention or correction) on three sets of antibiograms (A1-3) described in Table 1.

AST groups A1 and A2 consist of 571 and 74 antibiograms prepared during working routine in the microbiology laboratory of the University Hospital in Creteil, France. The samples were collected from patients of the hospital and the preparation and analysis of the AST was not designed primarily for our study but followed the normal hospital procedures. AST set A3 consists of eight Petri dishes prepared in the Hospital of Medecins Sans Frontieres in Amman, Jordan. In the case of this set, the plates were inoculated with microorganisms purchased from the American Type Culture Collection (ATCC) and routinely used for quality control. Such strains are among the main pathogens and have known inhibition diameters to various antibiotics. Species distribution, preparation information, and other details are reported in the Methods for all three sets.

**Data acquisition.** All plates in AST groups A1 and A2 were imaged with a smartphone camera (Honor 6x with a resolution of 12 megapixels). Since the App was still under development at the time these images were taken, we used the default Android camera application for acquisition. Then the images were analyzed with the App's full automatic procedure (without manual intervention). As control inhibition diameters for sets A1 and A2, we collected the measurements effectuated by the laboratory technician using a commercial automatic reading system (SIRscan[13], i2a, Montpellier, France). The control diameters measured by the technicians with the SIRscan system were extracted retrospectively from the hospital database, since these antibiograms were performed during routine analysis in the hospital. The SIRscan system allows for correction on automatic measurements. Nevertheless, for productivity reasons, the technician did not always adjust the diameters if the adjustment did not yield a different categorization result, therefore diameters can be unadjusted even if they give the right susceptibility categorization.

Among the pictures of AST groups A1 and A2, we selected standard and problematic pictures according to the following criterion: if for more than two antibiotics in the picture we found an absolute diameter difference between the App and control values of more than 20 mm, we considered the picture problematic, otherwise, it was considered standard. The problematic images are often associated with plates with defects or show very low inhibition-to-bacteria intensity contrast (due to low bacteria pigmentation and/or low illumination conditions). Nevertheless, most of the low contrast images in A1 and A2 were classified as standard (see Fig. 3).

All plates in A3 were imaged with the App on a smartphone (Samsung A10, 12 megapixels camera) by eight different laboratory technicians to take into account inter-operator variability (e.g., plate position, contrast, and random noise). The resulting 64 images were analyzed with the App's full automatic procedure. As a control, each AST was measured manually with a ruler by the same eight lab technicians. In this way, each inhibition diameter was measured eight times.

**Benchmark.** The diameters of the inhibition zones read with the App's automatic procedure were compared with the control diameters. For every diameter, we calculated the absolute difference with the corresponding control value. The susceptibility categorization (SIR) of the antibiotics was made for both the App's procedure and control, by comparing the inhibition diameter of each antibiotic to the breakpoint defined in the EUCAST guidelines[7]. Antibiotics for which a breakpoint was not provided are excluded. In order to evaluate the App's performance, we compared the susceptibility categorization of the automatic procedure to the control one. Following the same terminology proposed by[17,20,21], we calculated the agreement as the rate of identical categorization; disagreement is classified as very-major, major and minor. Very-major disagreement occurs when an antibiotic is categorized S (Susceptible) while the control is R (Resistant), major error corresponds to a categorization of R with control S and minor disagreement is any other categorization error involving the Intermediate value I (Intermediate). As a measure of agreement between the App and control, we calculated the unweighted Cohen's kappa index[25].

**Image processing performance.** The App's IP procedure proved its reliability at each step. For each dataset, the photos were automatically cropped to isolate the Petri dish. The automatic crop procedure never failed if the image respected the acquisition protocol (see Supplementary Note 1). The automatic pellet detection correctly found all antibiotic pellets in A1 and A2. In A3 0.5% of pellets were missed (false negatives). Half of the missed pellet show visible flaws (see Supplementary Fig. 4) and should not be considered in the analysis, according to experts' advice. False positives never occurred (other objects wrongly identified as pellets). In case a pellet was missed, the users can add it with the help of the graphical interface.

The antibiotic labels were always correctly interpreted in A3, even if the image was not perfectly focused or the text was damaged (by bad printing or by positioning it with tweezers). In datasets A1 and A2, the accuracy was 98%. Misclassification happened in cases of very poorly printed labels or for pellets from non-supported providers, on which the ML model was not trained. To overcome this problem, in the app we calculate a confidence value for each classification yielded by the model in order to reject misclassified labels and ask the user to identify them by eye (see Supplementary Note 3 for detailed results). Moreover, since the whole process is supervised by the user, misclassifications can easily be corrected.

Furthermore, we demonstrate a proof of concept for the ML classification of resistance mechanisms. We examined two clinically-relevant resistance mechanisms that are traditionally detected by the presence of non-circular inhibition zones. By training simple convolutional neural network models on the relevant portions of AST images, we obtained encouraging results: Accuracy higher than 99.7% in detecting induction (indicative of MLSb-inducible resistant Staphylococcus aureus) and 98% in detecting synergy indicative of ESBL production (see Supplementary Notes 5 and 6). However, due to user experience considerations in combination with concerns about model transferability, we ultimately determined not to incorporate these resistance mechanism-detecting models into the App. We do not exclude reconsidering this approach in a future version of the

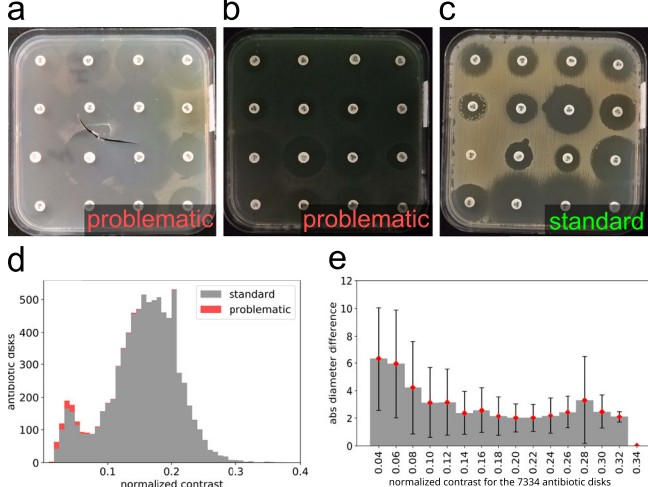

**Fig. 3 Problematic images and the role of intensity contrast.** A few problematic images have been identified in the datasets. These correspond to damaged plates (**a**) and images with very poor visible contrast between the bacteria and the inhibition (**b**). Some inhibition zones are hard to isolate, even by eye. For comparison, a standard image looks like (**c**). The coupled effect of bacteria pigmentation and variable illumination produces a considerable variability in the bacteria-to-inhibition intensity contrast (**a,b**, **c**). The histogram in (**d**) shows the distribution of image contrast for standard and problematic images in AST set A1 (the contrast is defined here as the difference between the central intensity level of bacteria and inhibition): problematic images (in red) are a small fraction of the total, mainly concentrated in the lower contrast region. Finally, **e** shows the observed mean diameter difference in millimeters versus contrast (Data are presented as mean values ± SD): low contrast images yield worse results.

**Table 2 Categorization agreement between the App automatic procedure and control. The number of antibiotics reported here is the number of those for which clinical breakpoints are provided by the EUCAST[7]. The lines of this table present the agreement/ disagreement for all antibiogram sets (A1, A2, and A3). For each line, we specify the control of diameter values, the number of analyzed images and corresponding antibiotic pellets, the agreement and disagreement (as defined in the text), the Cohen's Kappa coefficient as another measure of agreement. The label "overall" means that all pictures are considered, whereas standard and problematic stand for the respective images subsets section. The plates in sets A1 and A3 were grown on a standard Mueller–Hinton (MH) growth medium, whereas we used blood enriched MH in A2. In the last line of the table, the abbreviation av. stands for average. In this line, we used as control diameters the average value across the measurements of all eight technicians.**

| AST set | Control | Images | Antibiotics | Agreement (%) | Disagreement (%) | | | Kappa |
|---|---|---|---|---|---|---|---|---|
| | | | | | very major | major | minor | |
| A1 | | | | | | | | |
| overall | SIRscan | 570 | 7334 | 90 | 54 (0.7) | 428 (5.8) | 270 (3.7) | 0.77 |
| standard | SIRscan | 561 | 7223 | 90 | 46 (0.6) | 390 (5.4) | 269 (3.7) | 0.77 |
| problematic | SIRscan | 9 | 111 | 58 | 8 (7.2) | 38 (34.2) | 1 (0.9) | 0.12 |
| A2 | | | | | | | | |
| overall | SIRscan | 75 | 534 | 91 | 4 (0.7) | 36 (6.7) | 6 (1.1) | 0.71 |
| standard | SIRscan | 73 | 509 | 95 | 4 (0.8) | 16 (3.1) | 4 (0.8) | 0.83 |
| problematic | SIRscan | 2 | 25 | 12 | 0 (0) | 20 (80.0) | 2 (8.0) | 0.01 |
| A3 | | | | | | | | |
| overall | manual | 64 | 776 | 95 | 3 (0.4) | 7 (0.9) | 27 (3.5) | - |
| overall | manual (av.) | 64 | 97 | 98 | 1 (1.03) | 0 (0.00) | 1 (1.03) | 0.96 |

application if we can generalize it to a larger and diverse dataset. Instead, every time a resistance mechanism can appear in a culture (given the bacteria species and the tested antibiotics) the application will systematically ask the user to verify the presence of the associated shape, showing illustrated examples (see Fig. 2).

**Susceptibility categorization**. Our new diameter measurement approach yielded good classification results over most of the available images, as shown in Table 2. Only a small fraction of the pictures, classified as problematic in Table 2 (1.5%, 2.6%, and 0 in A1, A2, and A3, respectively), produced major discrepancies. Overall, diameter measurements allowed a susceptibility categorization agreement of at least 90% for all three antibiogram sets (all images included). The categorization results are reported in Table 2.

The actual distribution of the diameter differences among manual, automatic, and assisted (corrected by the user) readings of A3 are shown in Fig. 4. In general, we observe that the manual measurements (done with a ruler) are on average slightly larger than the automatic and assisted ones. The assisted measurements are done by the technicians directly on a smartphone. The graphical interface displays a circle centered on the antibiotic disk that the user can adjust in diameter until it fits the zone of inhibition. With this kind of visualization, the measurement is easier and more accurate than the one done with a segment (which is the case of the ruler). We argue that most of the diameter differences between automatic reading and control are due to the difficulty of measuring with a ruler the diameter of inhibition zones which are not perfectly circular (see Supplementary Fig. 4). Instead, more accurate measurements are obtained by adjusting a circular guide as in the App. The positive effect of measuring with the App is also visible in Fig. 4, which displays the inter-operator average diameter difference. As expected, we observe that using the App lowered inter-operator variability.

The choices we made in building the App and the acquisition setup are made in order to facilitate its adoption in the laboratory routine in low-resource settings. The counterpart of this choice is a certain difficulty in obtaining a constant image quality (notably because of the intrinsic variability of smartphone hardware and software). The acquisition setup and GUI assistance are there to limit this variability, but can not standardize the image acquisition at the level of commercial systems with dedicated hardware. Nevertheless, the data in this study display a certain variability (especially in contrast, as shown in Fig. 3) with which the application's IP could easily cope. Strong light reflections and other important issues will result in evident wrong readings, which are easily detectable by eye with the App's user interface. We remarked that, even without training, users can easily notice such problems and adapt the acquisition setup in order to eliminate them in future acquisitions.

Finally, we compared the App to other existing systems. The categorization agreement and errors observed in this studies among the App's automatic procedure and control are similar to those of other systems (free and commercial) found in the literature (see Supplementary Table 3). The image treatment has been designed to perform on mobile devices, and does not require the user's intervention to optimize image features (e.g., contrast). We obtained consistent results even by downscaling the antibiogram pictures up to a resolution of 1 megapixel (see Supplementary Table 4). The whole reading of one antibiogram (12 megapixels picture, 16 antibiotic disks) takes less than 1 s on a PC using one 2.3 GHz Intel Core i5 processor, 1.5 s on a high-end smartphone (Pixel 3 released in 2018), and 6.6 s on a low-end smartphone (Samsung A10), still much faster than manual reading.

For the matter of hardware compatibility, as of now, we have tested three smartphone models (Google Pixel 3A, Honor 6x, Samsung A10) ranging from high- to low-end. We thoroughly tested the most affordable and available model (Samsung A10) and can recommend it as a trusted device. In the future, we will maintain a list of recommended devices associated with the App.

**Discussion**
In this paper, we have presented a fully offline smartphone application capable of analyzing disk-diffusion antibiograms. The App assists the user in taking a picture of a disk diffusion AST plate, measuring and categorizing zones of inhibition, and interpreting the results. The analysis is performed entirely on the same device used to acquire the picture of the antibiogram.

The App shows performances similar to other existing automatic reading systems. In particular, the automatic inhibition zone diameter reading is consistent with manual reading (gold standard). The observed accuracy is therefore considered satisfactory for usage in an AST reading system assistant. A user-friendly interface makes it easy for the user to adjust the

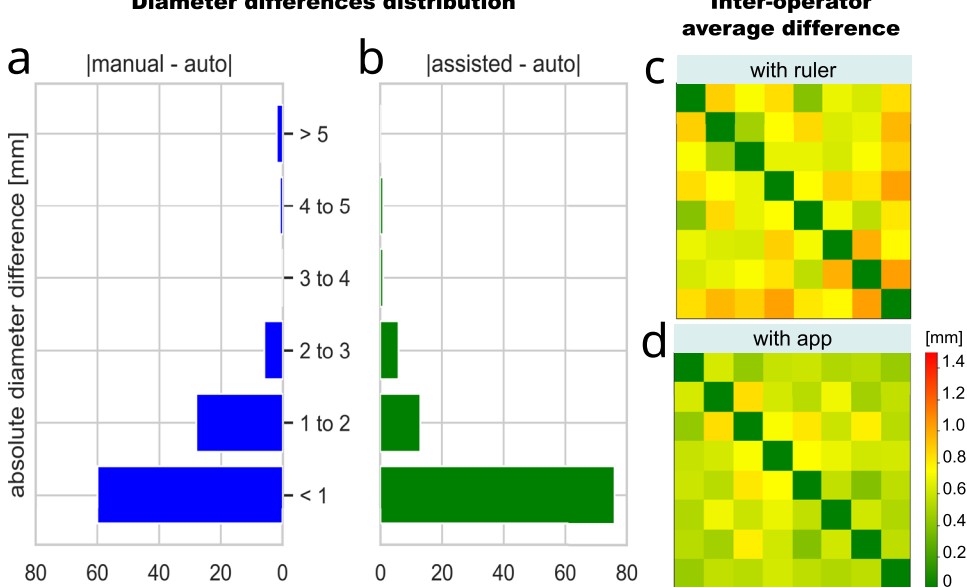

**Fig. 4 Benchmark results on dataset A3.** The histograms (**a,b**) show the distribution of the absolute diameter differences between the App's automatic procedure (auto) and the manual measurement with ruler (**a**) as well as with the diameter adjusted on the smartphone by the technicians (assisted, **b**). On the right, the heat-maps show the average absolute measurement difference among the eight technicians (given two readers *i* and *j*, square *i, j* represents the average difference between them) measuring with the ruler (**c**) and in assisted mode with the App (**d**). The assisted measure seems to reduce inter-operator variability.

automatic results if needed. We tested the App on antibiograms prepared with standard Mueller–Hinton (MH) growth medium as well as with MH supplemented with blood, used for fastidious organisms, and obtained similar results.

We built and trained two ML-based image classification models to identify resistance mechanisms. The accuracy results are encouraging, but given the relatively small training sets, we consider the risk of over-fitting too high for the scope of this mobile application. Nevertheless, these cases are handled by the integrated Expert System, which asks the user to confirm/exclude the presence of such shapes, when likely to happen.

The App aims to encourage the implementation of disk diffusion AST in resource-limited hospitals and laboratories where antibiograms are not routinely used or poorly interpreted. It does this by simplifying the measurement task and by providing an interpretation tool, offline, on a simple smartphone with a camera. The App is part of the mobile-Health[26] (mHealth) revolution, which aims to increase patients' access to testing, to aid in their treatment, and to decrease the digital gap in the world. Our hope is that the App could help fill the digital gap and increase patients' access to AST worldwide.

Further clinical investigations using the App in MSF hospitals will estimate the patient benefit enabled by AI-based antibiotic resistance testing. Pending the results, the mobile application will be released and open sourced to the public under the name of AntigbioGo. The App will support selective reporting of antibiotic sensitivity[27], an important component of the antibiotic stewardship strategy. It will offer the option of contributing data to global AMR surveillance with institutional bodies in place such as the WHO program GLASS (Global Antimicrobial Resistance Surveillance System) and/or WHONET in order to facilitate the collection of epidemiological data on antimicrobial resistance.

## Methods

**Detail of the AST sets**. In this section, we give a detailed description of the AST datasets used for the benchmark. The specific characteristics of each data set are summarized in Table 1. The bacterial species appearing in this study are reported in Fig. 5.

The plates were inoculated with 0.5 McFarland of a pure culture of the studied organism. Then antibiotic disks were positioned onto the plates with a dispenser gun (A1 and A2) or by hand (A3) and the plates were incubated from 16 to 24 h under aerobic or 5% $CO_2$ conditions depending on the species.

- Datasets A1 and A2: More than 91% of the plates in these datasets were square; the remaining 9% were circular. The antibiotic disks were bought from i2a (Montpellier, France) and positioned onto the plate with a dispenser gun. Antibiograms were performed according to the EUCAST[10] recommendations. Standard Mueller–Hinton agar was used in A1, whereas in A2, we used MH-F agar (blood-agar) for fastidious organisms (Biorad, Marnes-la-Coquette, France). The bacteriology laboratory of Creteil University Hospital is accredited under ISO15189 (Accreditation certificate N8-3372 rev. 9) therefore antibiotic disks and culture media are routinely quality checked.
- Dataset A3: This dataset consists of eight Petri dishes, all circular. The antibiotic pellets were produced by Liofilchem and positioned by hand with metallic tweezers. Specifically, the Petri dishes have been inoculated with the following ATCC dried microorganisms:

  *Pseudomonas aeruginosa* (ATCC 27583)
  *Klebsiella pneumoniae* carbapenemase producer (ATCC 700603)
  *Klebsiella pneumoniae* SHV-18-ESBL-producer (ATCC 700603)
  Fluoroquinolone susceptible *Escherichia coli* reference strains (ATCC 25922)
  Methicillin-Resistant *Staphylococcus aureus* (NCTC 12493)
  Vancomycin-sensitive *Enterococcus faecalis* (ATCC 29212)
  Gentamicin-resistant *Enterococcus faecalis* (ATCC 49532)

**Image processing procedure**. The IP module of the App consists of a custom C++ library and it is endowed with a Python wrapper module and a quick-start documentation. We have deliberately developed the IP as a standalone module in order to facilitate its use in other projects involving, for example, batch processing of many images, or the integration in a Desktop application with the development of dedicated image acquisition hardware. Our aim for the App is to keep the hardware and setup as simple as possible, which is why we adopted the smartphone strategy. Nevertheless, in other projects, the mobile phone could be replaced with a small cost device like a Raspberry Pi with a camera still using our IP library.

The first raw input to IP is an AST picture, which consists of a plate (Petri dish) to be cropped out from the remaining background (see Fig. 1). Cropping is accomplished with the *GrabCut* algorithm[28], with the assumption that the plate is approximately centered in the image (i.e., lies within the frame displayed on the camera screen). From the cropped plate image, we extract the dominant color, to distinguish the type of growth media (MH or blood enriched HM), and the shape of the plate (round or square). Finally, the image is converted to gray scale for further processing.

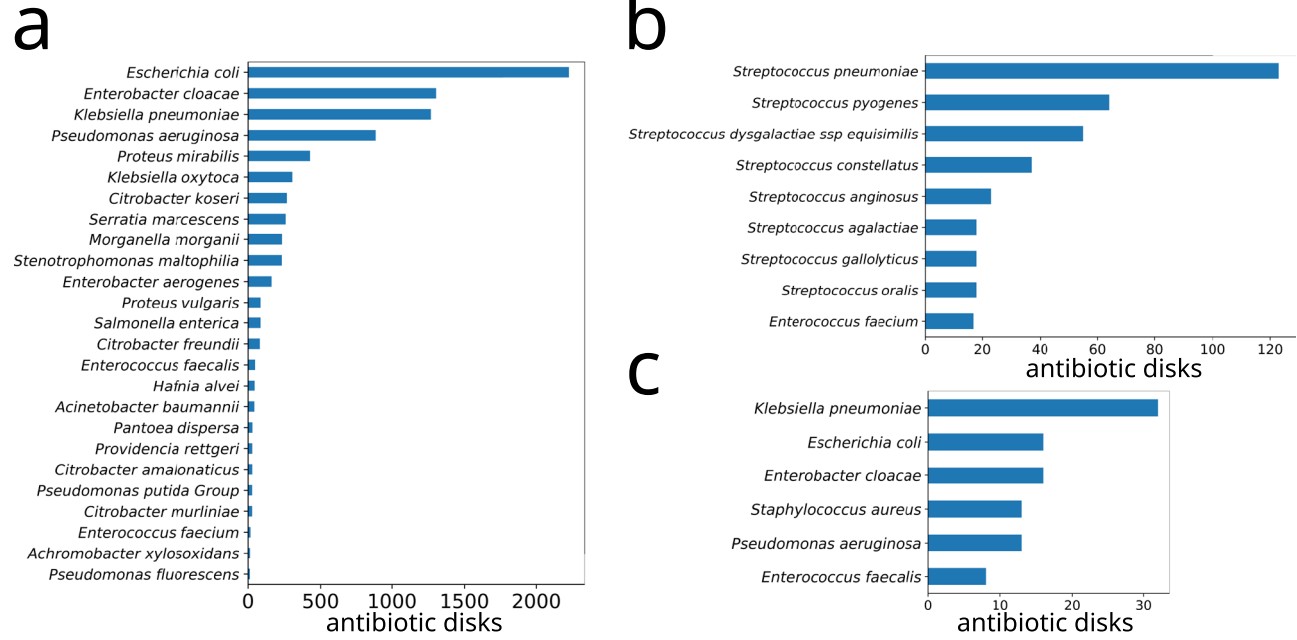

**Fig. 5 Distribution of species in the AST groups used in this study. a** Dataset A1. **b** Dataset A2. **c** Dataset A3.

The image of a plate (Fig. 1a) contains three main distinct components: the bacteria-free growth medium, the bacteria-covered growth medium, and the antibiotic disks. The latter are white round cellulose disks of known constant diameter (usually 6 mm). The precision of the whole automatic analysis depends on the accuracy of measuring their position and diameter in the image. Since the disk radius is known, the average disk radius in pixels is used to calculate the picture scale (pixel-to-mm ratio). Then, the inhibition diameters are measured from the disk center. The App features a fully-automatic method to measure antibiotic disks' positions and diameters based on intensity and shape, followed by user-assisted verification/correction. (see Supplementary Note 2).

Each pellet is printed with the acronym of the antibiotic it contains. There are only a few dozen antibiotics used in AST. Nevertheless, the acronym and the print features (font, shape, size, contrast, etc.) depend on the manufacturer of the antibiotic disks. The acronym of each antibiotic disk in the analyzed antibiogram must be read in order to retrieve the corresponding breakpoint for susceptibility categorization. Previously-proposed methods for reading these acronyms compared the image moment invariants[20] or used ORB (Oriented FAST and rotated BRIEF) descriptors[21,29]. For this task, we chose ML and trained a Convolutional Neural Network (CNN) model with Tensorflow[24] (see Supplementary Note 3 for details). We trained the model on a total of 18,000 images of antibiotic disks from two different manufacturers (resulting in 65 unique labels) and achieved 99.97% accuracy. In order to limit the out-of-distribution error (wrong classifications of disks on which the model was not trained), we used an ensemble of ten models and set a threshold on the output entropy. (see Supplementary Note 3 for details). The ML model showed to work also on poorly-printed disks and out-of-focus or low-resolution images. Interestingly we observed that even if the printed text is damaged when the disks are placed manually (using metal tweezers), the classification is always correct.

The inhibition zone diameter is the diameter of the largest circle centered on the antibiotic disk that does not include any bacteria; that is, the largest circle that can be drawn in the inhibition zone without touching any bacteria. In the easiest case, the inhibition forms a disk-shaped halo around the antibiotic pellet, but sometimes the disk is not well-defined, for example, because of the overlap of several inhibition zones or because the antibiotic disk lies close to the plate borders (see Figs. 6a, 7). The bacteria-to-inhibition intensity contrast in the image depends on the bacteria species. Also, illumination can vary both among different images and within the same image. The observable effect is a visible difference of contrast in the AST images (see Figs. 6a, 8, 9), especially when taken with a mobile phone where the illumination conditions cannot be controlled.

The new algorithm for automatic diameter measurement, presented here, is referenced as SWITCH (Spatial Weighted Intensity Threshold CHangepoint). SWITCH operates a $k$-means clustering of the pixel intensity locally (around each antibiotic pellet) to classify inhibition and bacteria pixels ($k = 2$). Successively, in order to find the inhibition zone boundary, it calculates and segments a radial profile $I(r)$ measured in the surroundings of the antibiotic disk (up to the closest neighboring disk). For each value of $r$ all pixels at distance $r$ from the pellet center are considered. The value of $I(r)$ is determined by the portion of pixels belonging to bacteria colonies (see details in Supplementary Note 4). Although SWITCH operates on a radial profile, the latter is calculated in a way that does not assume any preferential direction in the analysis of the image, which is important especially if the antibiotic disks are positioned by hand on the plate. Moreover, it partially takes into account the texture of the colony, thereby increasing robustness to noise.

**Susceptibility categorization and Interpretation.** In this study, the susceptibility categorization of the tested antibiotics (S/I/R) is done by comparing the measured inhibition zone diameters to the EUCAST clinical breakpoints[7]. The breakpoint values are stored offline in the application, within the expert system knowledge base. This base, which contains also the expert rules and other expert system resources, is maintained and updated yearly by i2a[13,15] (Montpelier, France).

In the context of AI, an Expert System is a program capable of taking reasoned conclusions from a given input, thereby simulating a human expert. Expert Systems have long been successfully used in microbiology[30] and most commercial systems use them today. An Expert System consists of an inference engine that takes reasoned conclusions on the input information, based on a set of rules written by human experts.

The Expert System integrated in the App takes as input the diameter of the inhibition zones of the observed plate. It categorizes the susceptibility of the bacteria to the tested antibiotics and runs a coherence check and a final interpretation. The coherence check examines the input information and alerts the user if incoherent data are found (for example, if an antibiotic is not coherent with the entered species, or if a natural resistance is not observed). The interpretation extrapolates the results to classes of antibiotics and produces final alerts for important resistance mechanisms.

**Auto-detection of resistance mechanisms.** Certain resistance mechanisms to antibiotics can be detected by disk diffusion AST because they often produce inhibition zones with characteristic shapes[31,32] (see Fig. 7). These shapes appear between specific antibiotic disks. If the disks are close enough, the antibiotic molecules they diffuse can interact and produce a synergy effect against the bacteria or induction of resistance. We used ML models to automatically recognize two particular shapes associated with two specific resistance mechanisms: synergy and induction, which can happen with *Extended-Spectrum β-lactamase (ESBL) production*[33] and *Macrolide-inducible resistance to Clindamycin*, respectively.

For each of the two tests, we trained a neural network model to classify positive vs. negative images. The models to recognize Clindamycin-inducible resistance and ESBL reach an accuracy of 99.7% and 98%, respectively (see Supplementary Notes 5 and 6 for details). However, the models have not been proven to perform well on a wide variety of images (varying in pellet arrangement, bacteria texture, etc). Also, since classification errors can have very serious consequences in AST interpretation and patient treatment, the App would need to ask the user for confirmation when automatically detecting a resistance mechanism. So, in the best case, including the resistance mechanism models in the App would bring only modest clinical improvements. Therefore, an ML-based automatic detection procedure was not included in the current version of the mobile application.

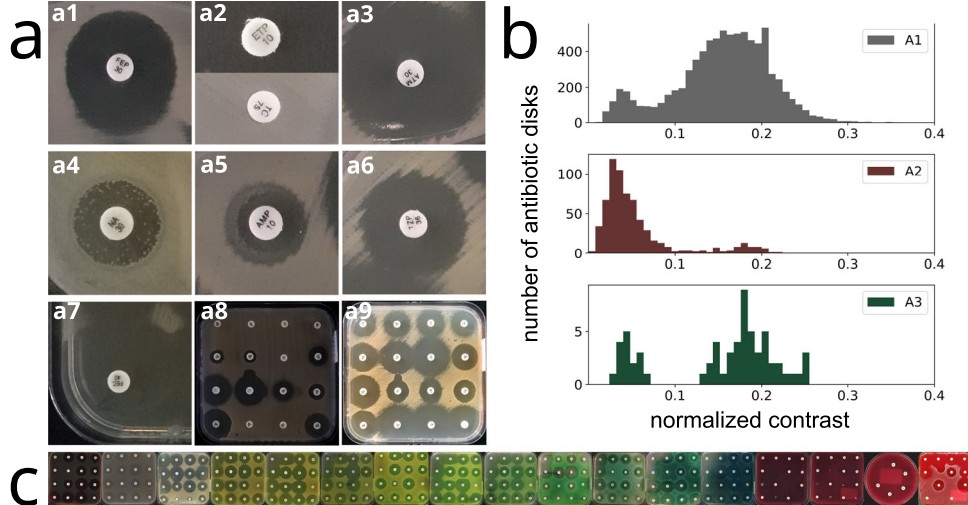

**Fig. 6 Variability in antibiogram pictures.** Examples of difficult cases for diameter reading (**a**). Non-circular inhibition shape (a1). Total or no inhibition (a1). Light reflections (a3). Colonies within the zone of inhibition (a4). Double inhibition zones (a5). Hazy borders (a6). Inhibition zone overlap and plate borders (a7). Low contrast (a8) defined as the difference between the inhibition and bacteria intensity value, compared to a high contrast (a9) image observed in dataset A1. The histograms in (**b**) show the contrast variability observed in the benchmark datasets (contrast is defined as the difference between the central gray levels of bacteria and inhibition, and normalized to the maximum available gray level). Observed variability in dominant hue (**c**).

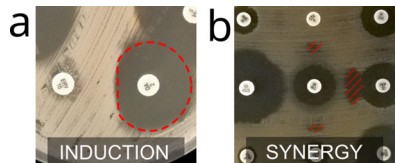

**Fig. 7 Characteristic shapes of inhibition zones due to resistance mechanisms. a** D-shape inhibition zone due to induction. **b** Appearance of inhibition between antibiotic disks due to synergy.

Nevertheless, the App asks users if they see such shapes wherever they are likely to appear, and shows them examples for comparison (Fig. 2).

**Reporting summary**. Further information on research design is available in the Nature Research Reporting Summary linked to this article.

## Data availability

The datasets analyzed during the current study are available online at http://stat.genopole.cnrs.fr/ast.zip. The data used for training the ESBL and D-SHAPE proof-of-principle models are available from The MSF Foundation upon reasonable request.

## Code availability

The image processing library described in this paper is distributed as open-source software at https://github.com/mpascucci/AST-image-processing. As of today, The App and its source code are available for research purposes upon request at https://form.typeform.com/to/qEGVBzbu. We plan to release the App as open-source software after approval of the CE authority as a clinical device. Fondation Medecins sans Frontieres sees the CE mark of its app solution (as a self-certified IVD SW) as a means to demonstrate and communicate on the quality and robustness of this digital tool for a non-profit. Waiting to comply to the IVDD Directive 98/79, they do not want to grant open access until they get this certification, as it could imply legal pursuit in France for the legal manufacturer (Fondation MSF) distributing a medical device without certification.

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

## Acknowledgements

The App is supported by: the Medecins Sans Frontieres Foundation, the Laboratoire de Mathematiques et Modelisation d'Evry, the Genoscope, the bacteriology laoratory of the Universitary Hospital Henri Mondor, Creteil France. For its potential impact, this project was awarded the 2019 Google AI Impact Challenge[34], selected among over 2600 applicants worldwide. We are especially thankful to Dominique Boissinot and Alain Jean of i2a for their fundamental help in the development of the Expert System. We thank all the organizers of the Google AI Impact Challenge and the people of Google.org, among others Anna Achilles, Sebastien Floodpage, and Mollie Javerbaum for their invaluable help.

## Author contributions

**Marco Pascucci** designed and coded the Image Processing system. He did the bibliographical research and wrote the manuscript. He did the preliminary feasibility study and built the first prototype of the application (selected in the Google AI impact challenge). He integrated the Expert System. He conceived the app's architecture. He designed the experiments concerning AST sets A1 and A2 and contributed to A3. He performed statistical analysis and interpreted the results. **Guilhem Royer** designed the experimental procedure for AST sets A1 and A2 and supervised data collection. He analyzed the data and wrote the paper. He designed the app and provided microbiology interpretation. He integrated the the Experte System. **Jakub Adamek** developed the mobile application, especially the UX, the integration of Tensorflow and tests. **Mai Al Asmar** and **Rasheed M. Fakhri** supervised the acquisition of the AST data in MSF Amman Hospital. **David Aristizabal** developed the mobile application. **Amine Bezzarga** developed the Expert System engine and contributed to the app development. **Laetitia Blanche** defined the user journey, the needs specification and coordination of the development, and release of the mobile application. **Guillaume Boniface-Chang** built and trained the model that recognizes the antibiotic pellets images. He participated to the writing of the manuscript draft. **Alex Brunner** did UX research. **Christian Curel**, provided the Expert System knowledge base. **Gabriel Dulac-Arnold** contributed in different aspects concerning ML. **Nada Malou** provided expert insights that proved to be critical for the overall definition of the project throughout all phases of development: mapping user journeys and pain points, identifying and prioritizing strategic requirements for the app, and enabling access to MSF on-field testing. She designed the experimental protocol concerning AST set A3. This protocol defined the workflow to collect pictures and manual measurements as well as underlying requirements. She also led on the submission of this protocol to MSF Ethical Review Board. She coordinated training and recruitment for the research assistants that performed the measurements in AST set A3. She tested the Expert System that was implemented as a core component of the app, by providing a representative set of complex interpretation cases. **Clara Nordon** took care that all the necessary conditions were reunited in order to ensure the conception and development of the app. **Vincent Runge** contributed to the algorithm for the automatic measurement of the inhibition diameters. **Franck Samson** contributed to the development of the mobile app. **Ellen Sebastian** contributed to the development of the mobile app. Ellen worked on the automatic recognition of the resistance mechanisms. She participated to the writing of the manuscript draft. **Dena Soukieh** designed the UX of the app. **Jean-Philippe Vert** contributed to the aspects concerning ML, discussed the results, and reviewed the manuscript. **Christophe Ambroise** designed and directed the study, interpreted the results and wrote the manuscript. **Mohammed-Amin Madoui** had the original idea of the application, designed and directed the study, interpreted the results and wrote the manuscript.

## Competing interests

The authors declare no competing interests.
