## [Peer Review File · Nature Communications]

Reviewers' Comments:

Reviewer #1:

Remarks to the Author:

In this paper, the authors present the first mobile application for automatically testing antibiotic resistance. The paper is well-written, and the authors tackle an important problem. If the application is finally openly released it can be useful for lots of researchers and hospitals; especially for those with low resources. In the literature, there are other open-source alternatives for automatically (or semi-automatically) test antibiotic resistance, but conducting the whole process in the save device can be a plus.

My main concern about this work is its open nature. The authors have not released the application, the code or the datasets employed in this work. Hence, it is not possible to test the application and try it. This is important not only for the review process but for the actual usage of the application. It is also important to release the code. The authors claim that the application works completely offline, but this kind of application might deal with sensitive information; so, it should be possible to audit the code. Finally, if possible the datasets employed in this work should be freely accessible, since this can help in the future to other researchers in the same area.

The authors also claim that the application is free, but they use an expert system that belongs to i2a; so, what is the license of this application? Do users will have to pay for updates of the EUCAST expert system?

Are the authors planning to release the application as a Desktop application? Working with a mobile phone might be a bit tedious since the screen is usually small; hence, the analysis of images might be a bit cumbersome; working with a tablet would help, but having a desktop application could be also useful. In addition, in the setting that the authors provide in Supplementary material S1, the mobile phone could be replace with a small cost device like a Raspberri Pi with a camera.

Apart from the aforementioned points; from my point of view, the paper is really well-written and adressed all the questions that came to my mind while I was reading it. I just have some additional minor comments.

- There is a webpage associated with the project (<https://mpascucci.github.io/ASTapp-overview>), but this is not mentioned by the authors up to Supplementary Section S1. In addition, they refer to a concrete page instead of the home page.

- The video is not provided as supplementary material.

- Is it possible to include CLSI rules?

- It should be quite simple to include an option to deal with antibiotic disks with a diameter different than 6mm.

- Include an explanation of the last row of Table 2 in its caption.

- The sentence of the first paragraph in Page 15 is incomplete.

- In the last paragraph of Supplementary material S3, the authors talk about test set, but before they only divided the set into training and validation. The authors should explain if the test set is the same as the validation set or if the dataset is split into three sets.

- Accuracy does not seem the best metric for detecting resistance mechanisms like induction or ESBL production since, as you explained in Supplementary material S5, the data is unbalanced;

this should be taken into consideration when making the folds (apply stratified folding) and also use other metrics like the F1-score. The same happens in Supplementary material S6.

- Taking into account that the analysis might contain sensitive information, are the authors planning to develop a webpage for researchers willing to release their data? As the author explain in the Introduction, the lack of data is one of the challenges in the fight against antimicrobial resistance.

- In which operating systems will the application run? Only in Android?

Some typos:

- Page 9: This base., -> This base,

- Line 4 of Supplementary material S3: sete -> set.

Signature:

Jónathan Heras Vicente

Department of Mathematics and Computer Science,
University of La Rioja, Spain.

Phone: (+34) 941299673

Email: jonathan.heras@unirioja.es

Web page: <https://www.unirioja.es/cu/joheras/>

Reviewer #2:

Remarks to the Author:

Pascucci et al have taken a big step towards facilitating DD on a global scale. The app they develop and which seems to be working well may play a pivotal role in the further dissemination and better access to the much used DD technology. My congratulations on a job well done. Then, unfortunately there is the document which I do not like much. I found the text confusing, the English of poor quality and there seems to be a constant mix between data, technology used and interpretation of results, irrespective of the section heading I am incapable of rewriting this (although I would very much like to!!)but there is a need for extensive editing. Below I will provide some generic remarks and questions and then below some examples of what I do not like in the text. The latter is incomplete. I would also suggest that for this paper the authors focus on the positive aspects and save all of the details and problems for a second paper. Key is to show the success you had, not to degrade that by including all sorts of complicated side steps.

GENERIC REMARKS

*Please explicitly state whether this is now good enough or whether you still need improvement of the technology. If the latter is the case then please provide your targets and what you need to do to attain these.

*During your studies have you been using any control strains? Just to make sure that the quality of the disks is conform expectations. As you know, quality differences between disks can be frustrating.

*Please make a statement of the (possible) usefulness of you app for other forms of diffusion testing (Etests).

*About resources and LMIC I think that there where DD is currently not feasible, the app is not going to add much. It will surely improve the quality of DD where it is being used already. So I do not know whether you will really improve patient access. You will improve quality and laboratory logistics for sure.

*In the methods section there is nearly nothing on how you did your DD and what methodology

was used. Please include a bit of microbiology here. In addition, there is very little discussion on reproducibility of your methodology. This needs a separate section in the text. I also find the description of the Expert System pretty superficial. What does it do? how does it do it? How good it is and what proof do you have for that?

MINOR ISSUES (INCOMPLETE)

Page 2, third section: I think you could easily delete this section since your readers will be familiar with the content.

Page 3, third section: Apparently image processing algorithms are not new. Make sure that you clearly define here what is new in your approach. Any patents to be mentioned?

Figure 1: In (a) the bacterial deck should not be mentioned a "colony". The legend is very sloppy (an instead of and, carboard instead of cardboard, what are "output interpreted results"?)

Page 5, line 5: The term "very close" is dangerous. What are you stating here? Is it good enough or just not? Better be precise here and use percentage or something to show in a more quantitative way what the differences are. Also, you may want to say something about adherence to CLSI or EUCAST guidelines here. And maybe even of FDA criteria of possible acceptance of the tool.

Figure 2: If there is a need for mandatory manual intervention (middle screenshot) then how automated is your solution?

Page 10, bottom section: Again, the "automatic" app needs user verification. What does that do to the term "automated"??

Page 11, Data acquisition: I do not understand the part where you state that because the app was still under development you decided to do something retrospectively. I have no problem with that, it is just unclear to me what exactly you did.

Page 13, line 10-11: Why do you have to state that if breakpoints are not available that the antibiotic involved was excluded. Why did you even test such antibiotics???

Page 14: Table 1 is at the heart of your paper. If you would explain this table in detail then the core of your paper were written. Some of the figures as illustration and job well done (see also my main comments above).

Page 14, first section "Results and Discussion": can be deleted 100% repetitive.

Page 17: Should be reduced in length by at least 50%

Reviewer #3:

Remarks to the Author:

The contribution is a mobile phone app - based on image processing and machine learning tools - capable to acquire images of Petri dishes prepared for antibiotic susceptibility testing (AST) and to provide an automatic reading of the plates. The authors claim that this first implementation of AST reading on a smartphone could contribute to increase access to AST especially in resource-limited settings and to positively contribute to the fight against antibiotic resistance.

I think the contribution can be considered relevant in terms of proof of concept but I feel some open issues remain to be addressed and I do not fully agree that, in this case, the fact that the method is implemented on a smartphone is of some value, at least not at the level stressed in the work. Anticipating my final suggestion, before entering in more details in some more specific observations, I find this work more adequate for a more application-oriented biomedical engineering or health informatics technical journal since I do not find the submitted contribution at a mature enough stage to be capable of concretely influencing way to think or to act in relation to the addressed problem.

Moreover, in terms of single adopted technologies, I did not find particularly impacting or highly novel ways to approach the acquisition and analysis of the images, while I found many potential disturbing factors that could negatively influence the performance and thus the reliability of the system. Due to this, but also according to how the system is conceived to work, high automation cannot be claimed since from what emerges by reading the paper, the presence of highly skilled

personnel able to control the acquisition setup, prevent/interpret anomalies, and supervise the whole process, is still necessary.

Some more specific appreciations/observations/concerns follow:

- I do not want to understate the main message of the paper, but since what proposed is not exactly an unconstrained "shot and read" mHealth system but requires a certain setup (illumination and power supply, acquisition stand, and a certain degree of standardization of the acquisition conditions), I find not strictly necessary, even taking into account economic factors, that the acquisition/analysis is made with a smartphone. Even another "smart" setup with a camera connected to a small PC or a small PC with a webcam attached to it could work. It is true that the combination of enough quality camera and modern computational power of today's smartphone is a favorable combination but I do not see this as an essential feature, at least not in the terms, a bit oversold, I found in the submission.

- In microbiology the current trend is automation and a high degree of standardization of the culturing procedures in order to concurrently increase quality and turnaround times (see e.g. <https://doi.org/10.1016/j.cmi.2019.11.008> that I would include in the references). I'm aware this cannot be granted all over the world and that low-income countries and/or decentralized health facilities are needing lightweight and smart solutions and that technology can help to alleviate the digital divide. However, automated AST interpretation is a tough problem in taken in its whole complexity and the impression I had is that the proposed solution, at the current stage, is still at the level of proof of concept, without having the strength of being a system genuinely deployable. In fact, there are many hypothetical sentences about the expected impact of the solution and many underestimated (or not fully addressed) issues that could intervene. In general, I found many points where is not clear what happens in case of malfunctioning. Among the main source of variability or critical aspects I can mention:

- o differences in optical and image quality settings among different smartphones;

- o potential variability in illumination conditions and poor description of illumination sources and configuration (proposed acquisition guidelines are too generic with respect to these highly critical aspects);

- o constant need of skilled technicians to supervise the acquisition and to interpret dubious cases contrasts with the sentence where authors say that third-party expert system is exploited to compensate lack of microbiology expertise. From what I read the proposed system cannot be used without the presence of experienced microbiology practitioners;

- o the grayscale conversion is a potential loss of information and the way is made is not fully justified;

- o user assistance is also necessary to correct errors in antibiotic discs location;

- o smartphone, antibiotic disc, agar plate manufacturers considered in the study are not enough representative of possible market variability and this constitutes a limit and in any case a source of uncertainty against the possibility of a direct and safe deployment of the technology in the target environments;

- o the method used to assess the diameter of the inhibition halo is based on a radial intensity profile extraction which is prone to every kind of geometric and radiometric errors potentially coming from acquisition and previous processing stages. I did not find any discussion nor experimental assessment about these issues.

- I appreciate the tentative to conceive a complete system able to address the task of AST interpretation, but I have to observe that constituent parts are either not novel or selected approaches are a bit outdated and/or not enough robust to handle the potential experimental variabilities. It is indicative, also in light of the concerns expressed in the above point, that potentially novel ML solutions to address more complex resistance behaviors only remains at a draft stage, and lack of any convincing experimental validation. An automated interpretation cannot overlook these aspects or just rely on the skillfulness of operators. In a high throughput lab, maybe suboptimal techniques could find a role in segregating more simple cases from ones that deserve the attention of highly specialized personnel, but for use in dispensaries or other small and decentralized facilities, I think that one should concentrate to solve the more complex

interpretation aspects rather than head at the value of using a smartphone with respect to other possible alternate low-cost setups (which could potentially guarantee an equivalent or even higher level of control and standardization of image acquisition).

All in all, an appreciable work but not at the level of quality I feel required for Nature Communications, at least at the current development stage. The system appears not capable to handle more complex and doubtful (but not infrequent) situations, without requiring skilled user monitoring and assistance. Therefore, the clinical improvement that this solution brings is not clear. The experimental validation is not complete and overlooks the level of complexity in real deployment scenarios. Moreover, I'm not sure the paper in its current structure is adequate for Nature Communications submission standards. The Method section is in the middle of the paper (while it should be quite independent at the end) and the long supplementary section seems to be an extra which I'm not sure it could be acceptably mapped without incurring in serious overlength issues.

Response to the reviewers

”The first AI-based mobile application for antibiotic resistance testing”

We are greatly thankful to the reviewers for their time and useful comments which helped us to refine the content of our manuscript make several important points clearer in the present version. Please find in this first section some general comments on our work which answer common points to the three Reviewers.

1 General response

1.1 Data and software availability

Reviewers demands on the code availability was highly legitimate and we considered to address them positively. The application is now available for research use only. Please find here some useful information for code access:

- Access to the Android version of the app should be asked via this form¹. After filling the form your request will be treated and you will be granted access to the Android application and its source code.
- The image processing module is already open source and publicly available at <https://github.com/mpascucci/AST-image-processing>. A quick-start documentation can be found at: <https://mpascucci.github.io/AST-image-processing>
- The benchmark data (images included) can be downloaded at: <http://stat.genopole.cnrs.fr/ast.zip>
- We have made available on YouTube a short video demo² of the application.

Please also note the following disclaimer of the MSF Foundation concerning the open-source nature of the application: *The application will be released as Open Source after approval of the CE authority as a clinical device. Fondation Médecins sans Frontières sees the CE mark of its app solution (as a self-certified IVD SW) as a means to demonstrate and communicate on the quality and robustness of this digital tool for a non-profit. Waiting to comply to the IVDD Directive 98/79, We do not want to grant open access until we get this certification, as it could imply legal pursuit in France for the legal manufacturer (La Fondation MSF) distributing a medical device without certification.*

1.2 The choice of smartphones

Reviewers wondered why we chose smartphone to support our application. Running the AST analysis completely on smartphones has key advantages for our use-case, namely an AST reading system for low-resource settings:

¹<https://form.typeform.com/to/qEGVBzbu>

²<https://youtu.be/0hNr9zTu6ig>

- zero additional hardware cost (smartphones are ubiquitous)
- compatibility: the smartphone OS deals with the hardware variability
- no continuous power supply needed (smartphone can work for hours on battery)
- everything we need is integrated in one device
- mobile interfaces are more intuitive and data is easier to manipulate visually.

The main drawback is probably the difficulty in standardizing the image acquisition quality. The images acquired with our protocol will never be as consistent as it is the case with commercial systems that use purposely developed hardware. Nevertheless, as far as we could see, using our app with the given acquisition protocol yields results which are in good agreement with manual measurement.

1.3 The application use case

When reading this paper, one should bear in mind that this application wants to help easing the adoption of disk diffusion AST by lowering the cost in specialized human resources, not to compete with highly sophisticated commercial systems. This is now clearly stated in the section “An all-in-one smartphone app for AST reading”.

Under this perspective, it is straightforward that full automation is not at all our aim. A clear and functional User Interface is at least as important as automation in our use-case. Automation, where present, helps standardizing the reading and make it easier, but we definitely want the technician to supervise the process since they can correct evident mistakes when they are shown on the application. Once again, we are not aiming to increase the antibiogram processing bandwidth of a high standard hospital.

1.4 The application development stage

Since the application is proven to be conform to the gold standard (i.e. manual measurement) it can be used as a measuring tool for inhibition zones diameters. The embedded expert system is created and updated by i2a. It is used in many high standard hospitals and laboratories and has proven its effectiveness in different scientific papers. At this stage, the application is distributed on demand for research use only, awaiting acceptance of the European Community as a medical device. Then it will be released freely and open-source.

2 Point-by-point response to the reviewers

Reviewer 1

In this paper, the authors present the first mobile application for automatically testing antibiotic resistance. The paper is well-written, and the authors tackle an important problem. If the application is finally openly released it can be useful for lots of researchers and hospitals; especially for those with low resources. In the literature, there

are other open-source alternatives for automatically (or semi-automatically) test antibiotic resistance, but conducting the whole process in the save device can be a plus.

My main concern about this work is its open nature. The authors have not released the application, the code or the datasets employed in this work. Hence, it is not possible to test the application and try it. This is important not only for the review process but for the actual usage of the application. It is also important to release the code. The authors claim that the application works completely offline, but this kind of application might deal with sensitive information; so, it should be possible to audit the code. Finally, if possible the datasets employed in this work should be freely accessible, since this can help in the future to other researchers in the same area.

Reply: We thank Reviewer 1 for the appreciations and the overall positive review. We had a small delay with the publication of the application, code and data because the application is undergoing the process of obtaining an official certification as a clinical device. See the “Common Points” section for details. Nevertheless access for research use and review can be now requested.

Reviewer Point P 1.1 — The authors also claim that the application is free, but they use an expert system that belongs to i2a; so, what is the license of this application? Do users will have to pay for updates of the EUCAST expert system?

Reply: The agreement between i2a and MSF states free access to the Expert System and its updates when used within the app in low-resource settings. The final licence will be published together with the public application.

Reviewer Point P 1.2 — Are the authors planning to release the application as a Desktop application? Working with a mobile phone might be a bit tedious since the screen is usually small; hence, the analysis of images might be a bit cumbersome; working with a tablet would help, but having a desktop application could be also useful. In addition, in the setting that the authors provide in Supplementary material S1, the mobile phone could be replaced with a small cost device like a Raspberry Pi with a camera.

Reply: No, we do not plan to release a Desktop application. The existing free solution AntibioGramJ is a valid alternative for this purpose. Nevertheless we developed the Image Processing library as a standalone module in order to facilitate its use in other projects (as it is now stated in Methods section 6.0.1). The idea of developing a small cost acquisition device is very interesting indeed. We evaluated it and finally decided to target smartphones because they are ubiquitous and ready to use.

Reviewer Point P 1.3 — Apart from the aforementioned points; from my point of view, the paper is really well-written and adressed all the questions that came to my mind while I was reading it. I just have some additional minor comments. There is a webpage associated with the project (<https://mpascucci.github.io/ASTapp-overview>), but this is not mentioned by the authors up to Supplementary Section S1. In addition, they refer to a concrete page instead of the home page.

Reply: The official page of the project is <https://fondation.msf.fr/projets/antibiogo>. We added a link in the paper in the main part, section “An all-in-one smartphone app for AST reading” as footnote. The concrete page mentioned in Supplementary is linked here because it shows the acquisition

setup and protocol. The information contained in that web page is complementary to the project's official page.

Reviewer Point P 1.4 — The video is not provided as supplementary material.

Reply: We had forgotten to upload the file. The demo video is now available on YouTube. Another (narrative) video is available at <https://fondation.msf.fr/en/projects/antibiogo>.

Reviewer Point P 1.5 — Is it possible to include CLSI rules?

Reply: Although the CLSI rules are not yet used in the actual version of the app, i2a has this rule set ready and it is in principle possible to use them. For the moment MSF has no interest in using CLSI rules but this decision may change in the future.

Reviewer Point P 1.6 — It should be quite simple to include an option to deal with antibiotic disks with a diameter different than 6mm.

Reply: Currently, we hard-coded the 6mm variable but an option could be included easily in the application's settings. Neither MSF nor its partner laboratory use pellets with a diameter different from 6mm, but it could be worth considering adding this parameter in the application's settings in future versions. Thank you for this practical advise.

Reviewer Point P 1.7 — Include an explanation of the last row of Table 2 in its caption.

Reply: We clarified this table element.

Reviewer Point P 1.8 — The sentence of the first paragraph in Page 15 is incomplete.

Reply: Indeed, this is now corrected.

Reviewer Point P 1.9 — In the last paragraph of Supplementary material S3, the authors talk about test set, but before they only divided the set into training and validation. The authors should explain if the test set is the same as the validation set or if the dataset is split into three sets.

Reply: The data is split in two. Test and validation sets are the same. We have removed this ambiguity from the supplementary section. Additionally, the benchmark images of antibiogram set A3 and the results reported in "Results and discussion" can be considered as a true test set for this model, because the model never saw them (directly or indirectly) during training.

Reviewer Point P 1.10 — Accuracy does not seem the best metric for detecting resistance mechanisms like induction or ESBL production since, as you explained in Supplementary material S5, the data is unbalanced; this should be taken into consideration when making the folds (apply stratified folding) and also use other metrics like the F1-score. The same happens in Supplementary material S6.

Reply: We have calculated F1 and AUC for both D-SHAPE and ESBL models and reported the results in the current manuscript version.

While in general we agree that accuracy is not the best measure when classes are unbalanced, we would like to point out that the values are so large for D-shape prediction that any other measure would result in similar conclusions. More precisely, when we report an average of 99.74% accuracy on 64 test images, this corresponds to an average error of 0.167 test images, i.e., a perfect classification for all test images in 83% of the train/test split experiments, and 1 mis-classified image in 17% of the train/test split. Any other measure (F1-score, balanced accuracy, AUC...) would therefore also be perfect in 83% of the train/test split, and be slightly lower because of one mis-classified image in 17% in the split; reporting other measures would therefore not bring additional information.

Reviewer Point P 1.11 — Taking into account that the analysis might contain sensitive information, are the authors planning to develop a web page for researchers willing to release their data? As the author explain in the Introduction, the lack of data is one of the challenges in the fight against antimicrobial resistance.

Reply: The benchmark data (images included) is now publicly available to download. We did not implement an image collection platform for the moment, but future versions of the application (which are already in development) will ask the user the consent to collect epidemiological data and share it with the WHO surveillance program. We are working with WHO to make our system dialog with WHOnet database.

Reviewer Point P 1.12 — In which operating systems will the application run? Only in Android?

Reply: This particular point has been a long debate among us. Although we started the development of the application for both operating systems, we are now considering focusing primarily on Android only, because the most affordable devices run this OS. We state now clearly that the application needs an Android smartphone in section 2.

Reviewer Point P 1.13 — Page 9: This base., – > This base, Line 4 of Supplementary material S3: sete – > set.

Reply: Thank you for noting these typos, which are now corrected.

Reviewer 2

Pascucci et al have taken a big step towards facilitating DD on a global scale. The app they develop and which seems to be working well may play a pivotal role in the further dissemination and better access to the much used DD technology. My congratulations on a job well done.

Reply: We are thankful to Reviewer 2 for appreciating the impact of such an application.

Then, unfortunately there is the document which I do not like much. I found the text confusing, the English of poor quality and there seems to be a constant mix between data, technology used and interpretation of results, irrespective of the section heading

.... I am incapable of rewriting this (although I would very much like to!!)but there is a need for extensive editing. Below I will provide some generic remarks and questions and then below some examples of what I do not like in the text. The latter is incomplete. I would also suggest that for this paper the authors focus on the positive aspects and save all of the details and problems for a second paper. Key is to show the success you had, not to degrade that by including all sorts of complicated side steps.

Reply: As reviewer 2 remarks, we do recognise that the form of the last submitted version of the paper did not clearly separate technology, biology and data. Most of the details have now been moved to the Methods section, leaving the most significant information in the main text. We are thankful to Reviewer 2 for having encouraged this through revision. In this reviewed version we have refined the structure of the text thereby significantly enhancing the readability.

Reviewer Point P 2.1 — Please explicitly state whether this is now good enough or whether you still need improvement of the technology. If the latter is the case then please provide your targets and what you need to do to attain these.

Reply: We have updated the Conclusions section, which now clearly states the state, availability and intended future developments of the application.

Reviewer Point P 2.2 — During your studies have you been using any control strains? Just to make sure that the quality of the disks is conform expectations. As you know, quality differences between disks can be frustrating.

Reply: Indeed, dealing with disks quality is an important matter. Fortunately, the bacteriology lab of CHU Mondor in Créteil is accredited under ISO15189 (Accreditation certificate N°8-3372 rév.9) therefore antibiotic disks and culture media are routinely quality checked. Thank you for asking, we have added this information in the Methods section of the manuscript.

Reviewer Point P 2.3 — Please make a statement of the (possible) usefulness of you app for other forms of diffusion testing (Etests).

Reply: For now, we only focus on diameter as other forms such as Etests are not currently performed on MSF's field because they are more expensive and require more stringent storage conditions (-20°C). Moreover Etest are really hard to read, as shown in this document, so it will require improvements in terms of shape detection.

Reviewer Point P 2.4 — About resources and LMIC I think that there where DD is currently not feasible, the app is not going to add much. It will surely improve the quality of DD where it is being used already. So I do not know whether you will really improve patient access. You will improve quality and laboratory logistics for sure.

Reply: This remark is really interesting, thank you. Indeed in the DD method, the antibiograms need good laboratory skills to be prepared at first. The need of expert microbiologists adds up to this entry cost and, as it is the experience of MSF, training technicians to prepare DD AST is relatively a minor problem compared to the availability of expert human resources needed to interpret the reading.

Therefore our application tackles the need of reducing the overall cost of, and therefore encouraging the implementation and use of DD AST in hospitals that do not use it yet.

Reviewer Point P 2.5 — In the methods section there is nearly nothing on how you did your DD and what methodology was used. Please include a bit of microbiology here. In addition, there is very little discussion on reproducibility of your methodology. This needs a separate section in the text.

Reply: True, we have now improved the methods section concerning the AST sets, we included details about the preparation of the ASTs.

Reviewer Point P 2.6 — I also find the description of the Expert System pretty superficial. What does it do? how does it do it? How good it is and what proof do you have for that?

Reply: We do understand the reviewer's concern about the expert system. The expert system used by our application is provided by i2a, a private company who employs it in its commercial devices. It was described and evaluated in specifically devoted papers (cited in our manuscript) such as: Hombach et al. "E. C. Standardisation of disk diffusion results for antibiotic susceptibility testing using the Sirscan automated zone reader. BMC Microbiology 13, 225 (2013). Medeiros et al. "Evaluation of the sirscan automated zone reader in a clinical microbiology laboratory. Journal of Clinical Microbiology 38, 1688–1693 (2000)

Nevertheless we have now added a deeper level of detail in the Methods' "Susceptibility categorization and Interpretation" section to better explain the expert system's functioning.

Reviewer Point P 2.7 — Page 2, third section: I think you could easily delete this section since your readers will be familiar with the content.

Reply: We expect readers to have various backgrounds, including computer science and image processing, therefore we believe that this section helps contextualize the text for such readers.

Reviewer Point P 2.8 — Page 3, third section: Apparently image processing algorithms are not new. Make sure that you clearly define here what is new in your approach. Any patents to be mentioned?

Reply: We thank Reviewer 3. Indeed the IP module uses a novel algorithm for the measurement of the inhibition diameters and the identification of the antibiotic disks, but it was not clear in the main text. In the current version of the manuscript this is stated clearly ("image processing" section).

Reviewer Point P 2.9 — Figure 1: In (a) the bacterial deck should not be mentioned a "colony". The legend is very sloppy (an instead of and, carboard instead of cardboard, what are "output interpreted results"?)

Reply: Indeed, we replaced "bacteria colony" with "bacteria" in the figure. The figure's legend is corrected and refined to better describe the pictures.

Reviewer Point P 2.10 — Page 5, line 5: The term "very close" is dangerous. What are you stating here? Is it good enough or just not? Better be precise here and use percentage or something to show in a more quantitative way what the differences are.

Reply: Thank you for this important remark. We replaced “very close” by “conform”. This paragraph being only a short overview of the paper’s plan, we present detailed quantitative results that support this claim in the Results and discussions section.

Reviewer Point P 2.11 — Also, you may want to say something about adherence to CLSI or EUCAST guidelines here. And maybe even of FDA criteria of possible acceptance of the tool.

Reply: We are sorry but we are not sure we understand entirely this remark of the Reviewer. The gold standard for AST diameter measurement is the manual reading for both EUCAST (cited in the paragraph) and CLSI guidelines. Concerning official acceptance, the application is undergoing the acceptance as clinical device in the European Community, but at this stages we think it would be premature to claim this in the paper.

Reviewer Point P 2.12 — Figure 2: If there is a need for mandatory manual intervention (middle screenshot) then how automated is your solution?

Reply: Our principal aim is not automation (see the first section on this document). Automation is there only to try to reduce inter-operator variability and make the reading process easier. In particular MSF noticed that technician can easily be thought to recognise those shapes if only they think they might be present on a plate: the application reminds them (automatically) to check, for those plates where the shapes could appear.

Reviewer Point P 2.13 — Page 10, bottom section: Againa, the ”automatic” app needs user verification. What does that do to the term ”automated”??

Reply: Following our previous answer, throughout the manuscript “Automatic” should be intended as “inferred automatically by the application and ready to be validated by the user”.

Reviewer Point P 2.14 — Page 11, Data acquisition: I do not understand the part where you state that because the app was still under development you decided to do something retrospectively. I have no problem with that, it is just unclear to me what exactly you did.

Reply: Retrospectively here refers only to the extraction of the diameter measurements from the hospital database. The rest of the sentence states that the picture were taken with the normal camera application of the smartphone (and not directly with our application). This part was indeed confusing. We hope the adjustments we did made it clearer.

Reviewer Point P 2.15 — Page 13, line 10-11: Why do you have to state that if breakpoints are not available that the antibiotic involved was excluded. Why did you even test such antibiotics???

Reply: In the bacteriology lab of CHU-Mondor in Créteil we use a restricted number of panels for routine testing. For example the same set of antibiotic disks are used when analyzing the susceptibility of *Pseudomonas aeruginosa* and *Stenotrophomonas maltophilia*. However EUCAST provides a breakpoint for the antibiotic Ciprofloxacin only for *P. aeruginosa*. So it is possible to compare the measure of the diameter of ciprofloxacin between the SIRscan and our app for both *P. aeruginosa* and *S. malto*, but the susceptibility S/I/R is available only for *P. aeruginosa*. This is the reason why we have more antibiotics in table 1 than in table 2.

Reviewer Point P 2.16 — Page 14: Table 1 is at the heart of your paper. If you would explain this table in detail then the core of your paper were written. Some of the figures as illustration and job well done (see also my main comments above).

Reply: The table is rather dense and, as Reviewer 2 observed, the former label was too succinct. In the current version of the manuscript the legend of Table1 has been refined and it meticulously describes the lines and column content. In general, all captions in the current version of the manuscript have been reviewed.

Reviewer Point P 2.17 — Page 14, first section "Results and Discussion": can be deleted 100% repetitive.

Reply: We agree with Reviewer 2. This paragraph has been removed and re adapted in the "Data acquisition" section which is now clearer. We thank the Reviewer for this useful hint.

Reviewer Point P 2.18 — Page 17: Should be reduced in length by at least 50%

Reply: Indeed, we have sensibly reduced the Conclusions section.

Reviewer 3

The contribution is a mobile phone app - based on image processing and machine learning tools - capable to acquire images of Petri dishes prepared for antibiotic susceptibility testing (AST) and to provide an automatic reading of the plates. The authors claim that this first implementation of AST reading on a smartphone could contribute to increase access to AST especially in resource-limited settings and to positively contribute to the fight against antibiotic resistance. I think the contribution can be considered relevant in terms of proof of concept but I feel some open issues remain to be addressed and I do not fully agree that, in this case, the fact that the method is implemented on a smartphone is of some value, at least not at the level stressed in the work. Anticipating my final suggestion, before entering in more details in some more specific observations, I find this work more adequate for a more application-oriented biomedical engineering or health informatics technical journal since I do not find the submitted contribution at a mature enough stage to be capable of concretely influencing way to think or to act in relation to the addressed problem.

Reply: We thank Reviewer 3 for appreciating the interest of our efforts. We kindly ask Reviewer 3 to re-evaluate this opinion about the impact of our application, under the perspective of the use-case presented at the beginning of this letter. Our application does not want to compete with highly automated commercial systems. Speed or productivity are not our main objectives at this stage. Our application wants to encourage the implementation and use of DD AST in resources-limited hospitals that do not use it yet. This is now clearly stated in the text (e.g. in the Conclusions).

Moreover, in terms of single adopted technologies, I did not find particularly impacting or highly novel ways to approach the acquisition and analysis of the images, while I found many potential disturbing factors that could negatively influence the performance and thus the reliability of the system. Due to this, but also according to how the system is conceived to work, high automation cannot be claimed since from what emerges by reading the paper, the presence of highly skilled personnel able to control the acquisition setup, prevent/interpret anomalies, and supervise the whole process, is still necessary.

Reply: Following the previous answer, according to us, the application uses enough technology to fulfill its task. The image processing engine introduced here has been designed to analyze images taken with smartphones and a very simple acquisition setups, hence adapted to the kind of image variability these loose conditions allow. The kind of errors that users can be asked to correct are essentially wrong readings of antibiotic labels or inhibition zone diameters, which, in our experience, are very easy to spot when represented on the application screen together with the actual picture, even if the technician is not an expert. Letting the users supervise the analysis is an a-priori choice more than a fallback strategy.

Reviewer Point P 3.1 — Some more specific appreciations/observations/concerns follow: I do not want to understate the main message of the paper, but since what proposed is not exactly an unconstrained “shot and read” mHealth system but requires a certain setup (illumination and power supply, acquisition stand, and a certain degree of standardization of the acquisition conditions), I find not strictly necessary, even taking into account economic factors, that the acquisition/analysis is made with a smartphone. Even another “smart” setup with a camera connected to a small PC or a small PC with a webcam attached to it could work. It is true that the combination of enough quality camera and modern computational power of today’s smartphone is a favorable combination but I do not see this as an essential feature, at least not in the terms, a bit oversold, I found in the submission.

Reply: This observation is very interesting and this has been a long debate among us before finally choosing the mobile application as our best option. We made the choice of using smartphones because of the long experience of MSF in limited-resource settings. We have interviewed laboratory technicians and doctors on MSF fields and considered the complex logistics there. The cost of shipping, installing and maintaining dedicated hardware was therefore considered discouraging for our purpose. The ubiquity of smartphones really is a point of great strength, even considering the shortcoming of losing hardware standardization.

Reviewer Point P 3.2 — In microbiology the current trend is automation and a high degree of standardization of the culturing procedures in order to concurrently increase quality and turnaround times (see e.g. <https://doi.org/10.1016/j.cmi.2019.11.008> that I would include in the references).

Reply: Indeed, this is effectively the current trends but only in high income countries. LMICs do not need to increase their turnaround times, they only need to be able to produce quality results. The use of this app is a step toward more standardization while offering ease of access and use. Thanks for the citation concerning an high standard commercial AST automatic reading system. We have added this to the paper in the introduction when introducing such systems.

Reviewer Point P 3.3 — I'm aware this cannot be granted all over the world and that low-income countries and/or decentralized health facilities are needing lightweight and smart solutions and that technology can help to alleviate the digital divide. However, automated AST interpretation is a tough problem in taken in its whole complexity and the impression I had is that the proposed solution, at the current stage, is still at the level of proof of concept, without having the strength of being a system genuinely deployable.

Reply: Indeed, our system is currently for research use-only and is undergoing further clinical studies to be accepted as a proper clinical device with all the needed warranties. However, the benchmark presented in our manuscript shows at least that the application can be used as a measuring tool, thereby reducing inter-operator variability associated with manual reading, which is also prone to distraction error.

Reviewer Point P 3.4 — In fact, there are many hypothetical sentences about the expected impact of the solution and many underestimated (or not fully addressed) issues that could intervene. In general, I found many points where is not clear what happens in case of malfunctioning.

Reply: We do not clearly understand what Reviewer 3 calls "malfunctioning". There might be two interpretations: A software error: in this case the application crashes without ending the analysis. Most of the partial results are saved and the analysis can be resumed later. These errors are very rare and we are doing all efforts to debug them in advance. A measurement error: The application's interface asks the users to verify every automatic result so that the user can adjust it if needed. The interface is clear enough to make the process of identification and correction very easy. Finally the Expert System runs a coherence check on the AST data before interpretation and alerts the user in case of incoherent or contradictory information, asking to check again.

Reviewer Point P 3.5 — Among the main source of variability or critical aspects I can mention: differences in optical and image quality settings among different smartphones;

Reply: We do agree: smartphone cameras are not made primarily for quantitative measurements. Nevertheless we noticed that: The 1mm AST reading sensibility is large enough for a smartphone camera to be precise. Plus, resolution is a minor issue: when degrading our images up to the resolution of 1megapixel, we could still find correct results. Most phones have nowadays a much higher resolution. We suggest an affordable and available device (Samsung A10) we have tested and intend to maintain a list of acknowledged devices. Also, we give an artificial AST image as a method to assess the distortion on the smartphone camera and if it is fit or not to be used with our app The application automatically uses the smartphone camera at its best quality

Reviewer Point P 3.6 — potential variability in illumination conditions and poor description of illumination sources and configuration (proposed acquisition guidelines are too generic with respect to these highly critical aspects);

Reply: The small acquisition setup we conceived is enough to allow a decent reading (only a few user interventions needed over several images). The image acquisition protocol although not very strict (for easier implementation), explains how to take pictures which are readable with the app in most of the cases. We noticed that the users understand by themselves how to adapt the conditions for an optimal reading, which reduces the need of their intervention. Overall the results of our work on images that

showed some variability (as displayed by the histograms of figure 3) suggests that the analysis of a smartphone camera image is possible and precise.

Reviewer Point P 3.7 — constant need of skilled technicians to supervise the acquisition and to interpret dubious cases contrasts with the sentence where authors say that third-party expert system is exploited to compensate lack of microbiology expertise. From what I read the proposed system cannot be used without the presence of experienced microbiology practitioners;

Reply: The Expert System acts as an “artificial microbiologist” in the sense that it allows the interpretation of the AST measurements in most of the cases.

Reviewer Point P 3.8 — the grayscale conversion is a potential loss of information and the way is made is not fully justified;

Reply: The grayscale conversion helps speeding up the image analysis (low-end smartphones are sometimes slow). The results show the accuracy, even with this conversion is high enough. Plus, colour is not a parameter mentioned in the EUCAST reading guidelines for manual reading.

Reviewer Point P 3.9 — user assistance is also necessary to correct errors in antibiotic discs location;

Reply: We are sorry Reviewer 3 understood this is the case. In reality user assistance is not needed for this. Antibiotics disks are automatically found at the right position. Only in the rare case of a miss, the user can add the missing antibiotic manually.

Reviewer Point P 3.10 — smartphone, antibiotic disc, agar plate manufacturers considered in the study are not enough representative of possible market variability and this constitutes a limit and in any case a source of uncertainty against the possibility of a direct and safe deployment of the technology in the target environments;

Reply: Reviewer 3 is completely right on this point. One of the strength of the high standard commercial system is their integration in this wild ecosystem of suppliers, even if some require their client to buy specific consumables to properly work (e.g. antibiotic disk). Some specific considerations: Our image processing algorithm is indifferent to the agar plate shape The app as it is now works well with two brands of antibiotic disks (those used by MSF). If a lab wants to use the app, changing antibiotic disk supplier is not difficult nor costly. Also, we plan to enlarge the support to other suppliers in future releases.

Reviewer Point P 3.11 — the method used to assess the diameter of the inhibition halo is based on a radial intensity profile extraction which is prone to every kind of geometric and radiometric errors potentially coming from acquisition and previous processing stages. I did not find any discussion nor experimental assessment about these issues.

Reply: To be more accurate, the proposed algorithm is not simply based on an intensity radial profile, but weights the pixels associated to bacteria and inhibition and takes into account the homogeneity of the profile. In this sense, it is more robust. Nevertheless we agree with Reviewer 3 and we consider the

remark valid for any measuring algorithm. For this very reason, we do minimal preprocessing on the images.

Reviewer Point P 3.12 — I appreciate the tentative to conceive a complete system able to address the task of AST interpretation, but I have to observe that constituent parts are either not novel or selected approaches are a bit outdated and/or not enough robust to handle the potential experimental variability.

Reply: The image processing algorithm proposed here is novel and was conceived especially for this app after studying alternatives found in the literature. We chose to implement a new library because we needed particular attention to the issue of image variability. A very key point is that our experiments proved its success. The app uses enough technology to process smartphone camera images with the simple setup we propose.

Reviewer Point P 3.13 — It is indicative, also in light of the concerns expressed in the above point, that potentially novel ML solutions to address more complex resistance behaviors only remains at a draft stage, and lack of any convincing experimental validation.

Reply: We investigated how ML could help solving our problem and indeed it showed to be very useful in reading the antibiotic disks labels (which is novel in literature). Concerning the classification of resistance mechanisms, as we stated, we showed that we were essentially limited by the number of training images. We also considered that further investments in this direction would be not worth it and its benefit would not counterbalance the risks. As explained in the text, this issue is currently tackled by the Expert System, which provides a viable solution for our target.

Reviewer Point P 3.14 — An automated interpretation cannot overlook these aspects or just rely on the skillfulness of operators. In a high throughput lab, maybe suboptimal techniques could find a role in segregating more simple cases from ones that deserve the attention of highly specialized personnel, but for use in dispensaries or other small and decentralized facilities, I think that one should concentrate to solve the more complex interpretation aspects rather than head at the value of using a smartphone with respect to other possible alternate low-cost setups (which could potentially guarantee an equivalent or even higher level of control and standardization of image acquisition).

Reply: This observation of Reviewer 3 is very interesting: the aspect of interpretation is delicate. Nevertheless MSF thinks the most important and difficult point in the implementation of AST in their hospital comes from the reading and interpretation, which are either imprecise or hard to teach to the lab technicians, therefore the conception of the application. Alternative low-cost solutions for disk diffusion AST reading cost 5-10k\$. Our approach is rather pragmatic. Our Application is a FREE tool destined to roughly formed technicians in difficult environments. Bringing an Expert System on a smartphone is a big step towards “solve the more complex interpretation aspects“. MSF Foundation develops another interesting project of 3D printed prosthetic. As you can imagine, these are not endowed with high technology and are far away from the standard of what rich countries' hospitals can afford. Nevertheless with this project MSF is satisfying the need of many who did not have access to prosthetic at all.

Reviewer Point P 3.15 — All in all, an appreciable work but not at the level of quality I feel required for Nature Communications, at least at the current development stage. The system

appears not capable to handle more complex and doubtful (but not infrequent) situations, without requiring skilled user monitoring and assistance. Therefore, the clinical improvement that this solution brings is not clear.

Reply: We hope the information we added here is enough to answer these concerns of Reviewer 3 at this point.

Reviewer Point P 3.16 — The experimental validation is not complete and overlooks the level of complexity in real deployment scenarios. Moreover, I'm not sure the paper in its current structure is adequate for Nature Communications submission standards. The Method section is in the middle of the paper (while it should be quite independent at the end) and the long supplementary section seems to be an extra which I'm not sure it could be acceptably mapped without incurring in serious overlength issues.

Reply: We understand the concerns of Reviewer 3. In fact, our experimental validation has been designed to be similar to other publications on the same subject (cited), concerning open or commercial AST reading systems. We have adapted the manuscript to better fit the style of the paper. The current version of the manuscripts has its Methods section at the end.

Reviewers' Comments:

Reviewer #1:

Remarks to the Author:

The authors have addressed all my concerns.

Reviewer #2:

Remarks to the Author:

I am fine with the revised manuscript. Useful new technology.

Reviewer #3:

Remarks to the Author:

With respect to the original submission, the revised one has improved paper structure and readability. Moreover, aims and limitations are more clearly evidenced. However, it is even more clear that the system is still at a proof-of-concept stage and its impact is not yet proven. This is because in-the-field validation is lacking so that there is still a too-wide gap between potential/desired impact and its scientific evidence. This does not invalidate the potential impact or deprive the value of the work and of the ideas behind it, but the scientific relevance is still too limited to justify publication in a high-end scientific journal as Nature Communications.

To evolve from a proof-of-concept stage either data-driven or usage-driven experiments (or even both) should be conducted in-the-field and should involve a variety of limited-resource contexts and a representative sample of technicians and specimens.

The experimental pieces of evidence which are currently provided are essentially based on data collected in controlled settings and are not surprising. Technologies used are standard and the paper is not so innovative from a technological point of view to justify a Nature journal publication.

I'm of course aware that full automation systems cannot be used as a benchmark in limited resource contexts, but I'm also more than aware that, while in standardized contexts many aspects are kept under control, partially automated AST image interpretation in a partially unstructured environment, where many variables are not under full control, can suffer from several problems and that would require one of the two (or even both): 1) advanced data-driven approaches (that I did not find in this work) based on large in-the-field data collection and capable to reliably handle all experimental variations, 2) human-centered user-experience impact-measurements experimentations based on various centers and involving various technicians. Probably the second scenario is more suitable for this App as it has been conceived and designed in order to prove that the proposed technology is enough to really promote the use of AST in context otherwise "resistant" to its use. However, the main impact-related proposed evidence that I derived from the paper is that a digital measurement of the halo diameter assisted by a graphical user interface can be more accurate than by the naked eye using a ruler, but this is quite self-evident, even independently from the presence/accuracy of an initial estimation.

What really needs to be measured is whether the technicians operating in-the-field are really helped by the technology as they must also assist it with a good degree of competence. It is necessary to be able to interpret this balance with all environmental parameters and turn them into impact evidence parameters in support of increased and more reliable use of AST in limited resource contexts. NatureComms requires evidence-based changing-mind contributions. I'm sorry but I do not see the possibility to change the mind and the way AST can be implemented in limited resource contexts without an in-depth in-the-field and human-in-the-loop experimentation. I still see this paper, at this stage, as better suited for a more application-oriented computer methods biomedical journal.

Some minor comments:

- I found not clear the criteria used to rate standard and problematic plates. A more detailed analysis of errors and causes that lead to disagreement should be given.
- The technical description of the SWITCH technique is not fully clear. More pictorial examples and more formalization should be given to justify the various steps and parameters selection. Not fully clear, why/how n_b (number of bacteria pixels) is measured in mm.

Response to the referees

Dear Reviewers,

By means of your careful review of our manuscript and interesting comments, we acknowledge a relevant improvement of the overall quality. Therefore we sincerely wish to thank you.

Best regards,

Marco Pascucci

Point-by-point

Reviewer #1 (Remarks to the Author):

The authors have addressed all my concerns.

Reviewer #2 (Remarks to the Author):

I am fine with the revised manuscript. Useful new technology.

We thank Reviewers 1 and 2 for their previous useful comments which led to this version of the manuscript, and also for their appreciation of the present version of the manuscript.

Reviewer #3 (Remarks to the Author):

With respect to the original submission, the revised one has improved paper structure and readability. Moreover, aims and limitations are more clearly evidenced. However, it is even more clear that the system is still at a proof-of-concept stage and its impact is not yet proven. This is because in-the-field validation is lacking so that there is still a too-wide gap between potential/desired impact and its scientific evidence. This does not invalidate the potential impact or deprive the value of the work and of the ideas behind it, but the scientific relevance is still too limited to justify publication in a high-end scientific journal as Nature Communications.

We thank Reviewer 3 for appreciating the improved readability of the manuscript and for underlining the impact of the proposed solution.

Having shown that the application's measuring precision is compatible with human reading, we have fulfilled the requirement for it to be used as an assistance tool, which, we think, is the first milestone for its adoption in the hospitals and laboratories. The application can be used as a measuring tool, which is fundamental for further clinical on-the-field investigation.

To evolve from a proof-of-concept stage either data-driven or usage-driven experiments (or even both) should be conducted in-the-field and should involve a variety of limited-resource contexts and a representative sample of technicians and specimens. The experimental pieces of evidence which are currently provided are essentially based on data collected in controlled settings and are not surprising. Technologies used are standard and the paper is not so innovative from a technological point of view to justify a Nature journal publication.

I'm of course aware that full automation systems cannot be used as a benchmark in limited resource contexts, but I'm also more than aware that, while in standardized contexts many aspects are kept under control, partially automated AST image interpretation in a partially unstructured environment, where many variables are not under full control, can suffer from several problems and that would require one of the two (or even both): 1) advanced data-driven approaches (that I did not find in this work) based on large in-the-field data collection and capable to reliably handle all experimental variations, 2) human-centered user-experience impact-measurements experimentations based on various centers and involving various technicians. Probably the second scenario is more suitable for this App as it has been conceived and designed in order to prove that the proposed technology is enough to really promote the use of AST in context otherwise "resistant" to its use.

We agree that further in-the-field experiments will be of absolute necessity for the development of new features in the application and its final distribution as a clinical tool. The results shown in this paper are essential to convince collaborating hospitals and laboratories to undergo such studies. All the components of the application might not be original (the image processing algorithm is novel though), but the orchestration and workflow is new, and yields an original and useful tool adapted to limited-resource settings, built together with professionals and technicians operating in these contexts.

However, the main impact-related proposed evidence that I derived from the paper is that a digital measurement of the halo diameter assisted by a graphical user interface can be more accurate than by the naked eye using a ruler, but this is quite self-evident, even independently from the presence/accuracy of an initial estimation.

We are glad that Reviewer 3 agrees with this point. In fact, this is not really self evident. Most of the interviewed technicians and even microbiologists did not even understand the limitations of measuring a circle with a ruler and made up personal rules of thumbs to measure a diameter of non circular inhibition zones. Still manual measurement with a ruler is the golden standard, presumably because an affordable universal alternative is not yet available. Maybe our application will indeed make a change.

What really needs to be measured is whether the technicians operating in-the-field are really helped by the technology as they must also assist it with a good degree of competence. It is necessary to be able to interpret this balance with all environmental parameters and turn them into impact evidence parameters in support of increased and more reliable use of AST in limited resource contexts. NatureComms requires evidence-based changing-mind contributions. I'm sorry but I do not see the possibility to change the mind and the way AST can be implemented in limited resource contexts without an in-depth in-the-field and human-in-the-loop experimentation. I still see this paper, at this stage, as better suited for a more application-oriented computer methods biomedical journal.

We do hope that our application will begin to change the way AST are used in limited resource context.

Some minor comments:

- *I found not clear the criteria used to rate standard and problematic plates. A more detailed analysis of errors and causes that lead to disagreement should be given.*

Standard and problematic plates are rated on an objective function of overall error magnitude. We have presented these two cases because we wanted to point out the problematic cases, hence the limitations, namely related to image contrast. The overall results are already good enough and we could avoid pointing out the problematic cases. We think that highlighting out the cases of major error in our dataset could help the users to easily improve the automatic results.

- *The technical description of the SWITCH technique is not fully clear. More pictorial examples and more formalization should be given to justify the various steps and parameters selection. Not fully clear, why/how n_b (number of bacteria pixels) is measured in mm.*

We thank Reviewer 3 for detecting a possible ambiguity here. With “ $n_b=1\text{mm}$ in the image scale” we mean that the value of n_b is chosen as the equivalent number of pixels representing 1mm in the image scale. The value of 1 mm is chosen because it corresponds to the requested precision of the diameter measurement. We have modified the text to state it clearly.